# PATRONUS: SAFEGUARDING TEXT-TO-IMAGE MODELS AGAINST WHITE-BOX ADVERSARIES

## ABSTRACT

Text-to-image (T2I) models, though exhibiting remarkable creativity in image generation, may be exploited to produce unsafe images. Existing safety measures, e.g., content moderation or model alignment, fail in the presence of white-box adversaries who know and can adjust model parameters, e.g., by fine-tuning. This paper presents a novel defensive framework, named PATRONUS, which equips T2I models with holistic protection to defend against white-box adversaries. Specifically, we design an internal moderator that decodes unsafe input features into zero vectors while ensuring the decoding performance of benign input features. Furthermore, we strengthen the model alignment with a carefully designed non-fine-tunable learning (NFTL) mechanism, ensuring the T2I model will not be compromised by malicious fine-tuning. We have conducted extensive experiments to validate the intactness of the performance on safe content generation and the effectiveness to reject unsafe content generation. Experiment results have confirmed the resistance of PATRONUS against various fine-tuning attacks by white-box adversaries.

## 1 INTRODUCTION

Text-to-image (T2I) models (Rombach et al., 2022; LMU, b; mid; Inc.) dazzle us with their stunning performance and amazing creativity. However, ethical issues with T2I models regarding unsafe content generation, like sexual-explicit, violent, and political images (Williams, 2023; Milmo, 2023; McQueen, 2023; Hunter, 2023a;b), are also alarming. An unprotected T2I model can easily be prompted to generate a large number of unsafe images. The Internet Watch Foundation discovered that countless images of child sexual abuse produced by T2I models had been distributed on the dark web (Milmo, 2023), causing potential sexual exploitation and sexual abuse (McQueen, 2023; Hunter, 2023a;b). Therefore, shielding T2I models from being exploited for unsafe image generation has significant research implications.

Existing defenses can be classified into two categories, *i.e.*, content moderation (Li; LMU, a) and model alignment (Schramowski et al., 2023; Gandikota et al., 2023). Content moderation aims to detect and block unsafe input prompts (Li) or output images (LMU, a). This is achieved mainly by input filters, output filters, or both. However, the filters are usually external to the T2I model and can be easily removed by white-box adversaries at the code level (Reddit, 2022). Model alignment aims to fine-tune the T2I model to eliminate its learned unsafe concept (Schramowski et al., 2023; Gandikota et al., 2023). Though being internally resistant to unsafe content generation, safely-aligned models are easily corrupted by fine-tuning with a small number of unsafe images.

In this paper, we propose PATRONUS, a defensive framework that strengthens the diffusion and decoder modules of a pre-trained T2I model. The design goal of PATRONUS is three-fold. (1) *Rejection of unsafe content generation*. The protected model should refuse to output unsafe content. (2) *Resistance to malicious fine-tuning*. The protected model should refuse to output unsafe content even if the model is fine-tuned with unsafe samples. (3) *Intact performance of benign content*. The protected model should preserve the performance regarding benign content. The workflow of PATRONUS is illustrated in Figure 1.

**Rejection of unsafe content generation**. Compared with input moderation, output moderation does not depend on input prompts and is more generalizable to unseen malicious prompts. Therefore, we devise a conditional decoder, which decodes only benign generations from the diffusion module (i.e.,

Figure 1: The objective of PATRONUS. 1) Inseparable moderation that defeats the adversary's detaching process. 2) Non-fine-tunable safety mechanism that defeats the adversary's malicious fine-tuning.

processing of the input) but refuses unsafe ones. This conditional decoder is inseparable from the T2I model. We achieve this through a prompt-independent fine-tuning on the decoder. Specifically, we input unsafe images into the encoder to collect unsafe features and then direct the decoder's decoding of these features to zero vectors. In this way, we assure the generalizability of the decoder's defensive capabilities against unsafe features.

**Resistance to malicious fine-tuning**. White-box adversaries may use a diversity of fine-tuning techniques to corrupt the moderated T2I model. Inspired by the idea of adversarial training, we align the moderated model through a $\text{min-max}$ game, simulating a worst-case adversary who attempts to regain unsafe content generation with malicious fine-tuning. The $\text{min}$ optimization mimics the adversarial goal of decreasing the performance loss on unsafe samples, and the $\text{max}$ optimization aims to suppress the fine-tuned performance obtained in the $\text{min}$ optimization. To achieve generalization, we construct a bag of fine-tuning strategies, including different optimizers, learning rates, iterations, batch sizes, and training sizes. Through a mixed sampling of fine-tuning strategies, the model's robustness and generalization to different fine-tuning processes are improved.

**Intact performance of benign content**. The performance of benign prompts may be degraded during the model alignment process. To tackle this difficulty, we utilize multi-task learning to achieve a balance between the performance of safe content generation and the resistance to unsafe content generation by adaptively computing appropriate weighting coefficients for these two objectives.

Extensive experiments have been conducted to evaluate the performance of PATRONUS. For I2P and SneakyPrompt datasets, PATRONUS can maintain the CLIP score of unsafe prompts to as low as 16.5 (visually black images) even after 500 malicious fine-tuning iterations. We demonstrate that it is hard to allure our protected model to produce unsafe content using any trick, and the cost of instigating an attack on our model is relatively high. We will open-source our code in the hope of incentivizing more research in the field of AI ethics.

We summarize our theoretical and technical contributions as follows:

- We make the pioneer attempt to investigate and validate the feasibility of a defense against white-box adversaries for T2I models. We innovatively apply the concept of non-fine-tunable learning to the T2I scenario.

- We design an inseparable content moderation mechanism that is prompt-independent. Additionally, our approach can resist malicious fine-tuning within a given budget, imposing significant costs on the adversary.

- We conduct extensive experiments to verify the effectiveness and robustness of PATRONUS against a diversity of adversarial prompts and malicious fine-tuning strategies.

## 2 BACKGROUND

In this section, we briefly discuss the preliminary knowledge of T2I generation and related work necessary for illustrating our method. Two lines of defenses are relevant to our study: 1) content moderation and 2) model alignment.

**T2I pipeline:** Consider a T2I pipeline, parameterized by $\theta$ (noted as $\mathcal{M}_\theta$), it involves three cascading modules, text encoder $\mathcal{M}_{enc}$, diffusion module $\mathcal{M}_{diff}$, and decoder $\mathcal{M}_{dec}$, *i.e.*,

$$\mathcal{M}_\theta = \mathcal{M}_{dec} \circ \mathcal{M}_{diff} \circ \mathcal{M}_{enc}. \tag{1}$$

Let $x^t$ represent the textual prompt. The text encoder takes $x^t$ as input and results in a conditioning vector. Then, the diffusion module generates a low-resolution feature with the guidance of the conditioning vector and participation of noise sampled from the Gaussian distribution. Finally, the decoder reconstructs the diffusion feature back to the original pixel space, *i.e.*, high-resolution images.

**Content Moderation:** There are two types of content moderators: input filters and output filters. Input filters are applied before the text encoder, detecting whether the textual prompt contains unsafe words (Jieli). A T2I equipped with the input filter $\mathcal{F}_i : \mathbb{T}^d :$ texual space $\to \mathbb{Y} = \{0, 1\}$.

$$\mathcal{M}'_\theta = \mathcal{M}_\theta \circ \mathcal{F}_i = \mathcal{M}_{dec} \circ \mathcal{M}_{diff} \circ \mathcal{M}_{enc} \circ \mathcal{F}_i. \tag{2}$$

$$\mathcal{M}'_\theta \left(x^t\right) = \begin{cases} \varnothing & \text{if } \mathcal{F}_i \left(x^t\right) = 1, \\ \mathcal{M}_\theta \left(x^t\right) & \text{if } \mathcal{F}_i \left(x^t\right) = 0. \end{cases}$$

Where $\mathcal{F}_i \left(x^t\right) = 1$ (or 0) signifies that the moderator regards $x^t$ contains (or does not contain) unsafe content. However, the input filters can be easily bypassed by adversarial prompts (Yang et al., 2023), which fulfill the goals as follows,

$$\mathcal{F}_i \left(x^t\right) = 1, \quad \mathcal{F}_i \left(\hat{x}^t\right) = 0, \quad \mathcal{M}_\theta \left(x^t\right) \approx \mathcal{M}_\theta \left(\hat{x}^t\right). \tag{3}$$

where $x^t, \hat{x}^t$ represent the original unsafe prompt and the corresponding adversarial prompt, respectively.

Output filters, $\mathcal{F}_o : \mathbb{R}^{H \times W \times C} \to \mathbb{Y} = \{0, 1\}$, can circumvent this issue, enabling more precise generation moderation since they directly review the compliance of the generated images as

$$\mathcal{M}'_\theta = \mathcal{F}_o \circ \mathcal{M}_\theta = \mathcal{F}_o \circ \mathcal{M}_{dec} \circ \mathcal{M}_{diff} \circ \mathcal{M}_{enc}. \tag{4}$$

$$\mathcal{M}'_\theta \left(x^t\right) = \begin{cases} \varnothing & \text{if } \mathcal{F}_o \left(\mathcal{M}_\theta \left(x^t\right)\right) = 1, \\ \mathcal{M}_\theta \left(x^t\right) & \text{if } \mathcal{F}_o \left(\mathcal{M}_\theta \left(x^t\right)\right) = 0. \end{cases}$$

Where $\mathcal{F}_o \left(\mathcal{M}_\theta \left(x^t\right)\right) = 1$ (or 0) signifies that the moderator regards the output contains (or does not contain) unsafe content. However, output filters cannot be applied to defend the white-box adversary due to its structurally separable nature, *i.e.*, the adversary can directly detach $\mathcal{F}_o$ from $\mathcal{M}'_\theta$ at the code level.

**Model alignment:** Model alignment family fine-tunes the diffusion module $\mathcal{M}_{diff}$, parameterized by $\theta_{diff}$, to improve compliance. SLD (Schramowski et al., 2023) and ESD (Gandikota et al., 2023) aim to guide the diffusion module away from unsafe regions or suppress harmful concepts during the denoising process, like:

$$\theta^*_{diff} = \arg \min_{\theta_{diff}} \sum_{x^t \in \mathbb{U}} -log(\mathcal{M}_{diff}(\mathcal{M}_{enc}(x^t), z), \tag{5}$$

where $\mathbb{U}$ represents the unsafe prompts. However, they rely on predefined prompts to participate in training or inference to some extent, which means the generalization cannot be guaranteed. Safe-Gen (Li et al., 2024) identifies the significance of vision-only layers (parameterized by $\phi$) to achieve text-agnostic mitigation and fine-tunes diffusion parameters against adaptive attackers.

$$\phi^* = \arg \min_{\phi \subset \theta_{diff}} \sum_{x^t \in \mathbb{U}} -log(\mathcal{M}_{diff}(\mathcal{M}_{enc}(x^t), z), \tag{6}$$

Compared with external filters, these methods encode the defensive property into the existing parameters. However, we find that their defensive performance can be easily corrupted by fine-tuning with only a dozen unsafe data and iterations.

## 3 DESIGN GOAL

In this part, we lay out three major goals of PATRONUS. Let $p_u, p_b$ represent unsafe prompts and benign prompts.

**Goal I: Rejection of Unsafe Content.** The model should refrain from generating images that contain visible malicious semantic information when confronted with unsafe prompts, *i.e.*, $\mathcal{M}_\theta(p_u) = \varnothing$. $\varnothing$ represents the absence of unsafe concepts, same hereafter.

**Goal II: Resistance to Malicious Fine-tuning.** Even after being fine-tuned with unsafe data by an adversary, the model should still be unable to generate images that contain visible unsafe content, *i.e.*, $\phi(\mathcal{M}_\theta)(p_u) = \varnothing$.

**Goal III: Preservation of Normal Performance.** The model should maintain similar outputs to the original model when presented with benign prompts, *i.e.*, $\mathcal{M}_\theta(p_b) \approx \mathcal{M}_0(p_b)$.

To integrate these goals in a unified framework, we formally formulate PATRONUS as follows,

$$
\begin{aligned}
\min_\theta \ & \mathbb{E}_{p\sim\mathcal{D}_m,\phi\sim\Phi}\ \mathcal{S}\left(p, \phi\left(\mathcal{M}_\theta\right)\right), \\
\text{s.t.}\ & \mathbb{E}_{p\sim\mathcal{D}_b}\left(\max\left\{0, \mathcal{S}\left(p, \mathcal{M}_0\right) - \mathcal{S}\left(p, \mathcal{M}_\theta\right)\right\}\right) < \epsilon,
\end{aligned}
\tag{7}
$$

where $\mathcal{D}_m$, $\mathcal{D}_b$, $\Phi$ represents the distribution of unsafe prompts, benign prompts, and fine-tuning processes, respectively. Note that $\Phi$ contains the case where the adversary does not fine-tune and directly prompts. $\mathcal{S}$ is a measure used to assess the quality of generated images, *e.g.,* the CLIP score (Radford et al., 2021). $\epsilon$ is the tolerance of the performance degradation on benign prompts. Since the constrained optimization problem in Equation 7 is difficult to solve, we introduce the Lagrange multiplier and solve the corresponding unconstrained optimization problem as

$$
\min_\theta \mathbb{E}_{p\sim\mathcal{D}_m,\phi\sim\Phi}\ \mathcal{S}\left(p, \phi\left(\mathcal{M}_\theta\right)\right) - \lambda \cdot \mathbb{E}_{p\sim\mathcal{D}_b}\left(\mathcal{S}\left(p, \mathcal{M}_\theta\right)\right).
\tag{8}
$$

In the rest of the paper, we provide an implementation of PATRONUS that can effectively solve the formulation objective.

## 4 METHODOLOGY

### 4.1 OVERVIEW

**Key Idea:** To tackle the limitations of existing defenses, which can be structurally removed or disrupted by malicious fine-tuning, we develop PATRONUS. The intuition of PATRONUS contains three aspects: 1) To achieve structurally inseparable output moderation, we embed the output filter within the decoder. Specifically, we govern the decoder module, *i.e.*, making it perform conditional decoding based on the safety of the generated features. 2) To ensure the defensive performance survives the adversary's fine-tuning, we enhance the defended components, including the conditional decoder and the aligned diffusion, with non-fine-tunability. 3) We carefully control the defense process to exclude the involvement of attack prompts, achieving prompt-independent defense, which guarantees robustness against various unsafe prompts.

**Overall Pipeline:** Starting from a pre-trained T2I pipeline, we implement PATRONUS by fine-tuning the decoder and the diffusion modules. First, we fine-tune a conditional decoder, which refuses to decode unsafe features, to achieve an inseparable moderator. Then, we create a non-fine-tunable safety mechanism to enable the conditional decoder and the aligned U-Net to resist malicious fine-tuning. Additionally, we pay attention to benign performance preservation that encourages the model to review the knowledge of benign inputs. Figure 2 describes the pipeline of PATRONUS. We summarize the overall process of PATRONUS in Algorithm 1.

### 4.2 INSEPARABLE MODERATOR

In this section, we design and realize an inseparable moderator by fine-tuning a decoder that performs the conditional output based on the feature's safety. Equivalent to having an output moderator $\mathcal{F}_{emb}$ embedded internally, the conditional decoder can be formalized as

$$
\mathcal{M}'_{dec} = \mathcal{M}_{dec} \odot \mathcal{F}_{emb}.
\tag{9}
$$

Figure 2: Design of PATRONUS. PATRONUS mainly consists of two processes, *i.e.*, the fine-tuning a (FTS) loops and the normal training reinforcement (NTR) loops. The FTS loops are designed to simulate different fine-tuning processes and degrade the fine-tuning performance in the restricted domain. The NTR loops are designed to maintain the performance in the original domain. The number of tasks $N$, the number of updates $K$, the learning rates of FTS loops $\alpha$ and NTR loops $\beta$, and the number of FTS loops $\ell_{\text{FTS}}$ and NTR loops $\ell_{\text{NTR}}$, and the total number of iterations Iter are hyper-parameters.

$$\mathcal{M}'_{dec}\left(f^t\right) = \begin{cases} \varnothing & \text{if} \mathcal{F}_{emb}\left(f^t\right) = \texttt{False}, \\ \mathcal{M}_{dec}\left(f^t\right) & \text{if } \mathcal{F}_{emb}\left(f^t\right) = \texttt{True}. \end{cases}$$

Where $\mathcal{F}_{emb}\left(f^t\right) = \texttt{True}$ (or $\texttt{False}$) signifies that the moderator regards $f^t$ as a safe (or unsafe) feature. $\odot$ denotes the $\texttt{AND}$ operation. $f^t$ is the feature generated by the diffusion corresponding to textual input $x^t$, obtained by

$$f^t = \left(\mathcal{M}_{diff} \circ \mathcal{M}_{enc}\right)\left(x^t\right). \tag{10}$$

Developing such a conditional decoder is far from trivial. We achieve it by fine-tuning the pre-trained decoder with the combined loss from two processes, *i.e.*, the conditional decoding process, and the feature calibration process as

$$\mathcal{L}_{\text{im}} = \alpha \cdot \mathcal{L}_{\text{cd}} + \beta \cdot \mathcal{L}_{\text{fsc}}. \tag{11}$$

### 4.2.1 CONDITIONAL DECODING

We perform an image-oriented alignment to direct the decoder to decode unsafe features into zero vectors while ensuring its decoding behaviors for benign features remain intact, which we describe as conditional decoding. Given the pre-trained VAE, consisting of a encoder $\mathcal{E}$ parameterized by $\phi$ and a decoder $\mathcal{D}$ parameterized by $\theta$. They were trained with the objective function as

$$\mathcal{L}(\theta, \phi) = -\mathbb{E}_{z \sim q_\phi(z|x)}[\log p_\theta(x|z)] + \text{KL}(q_\phi(z|x)||p(z)). \tag{12}$$

Optimizing the first loss brings $\mathcal{D}$ the conditional generation capability, opening up the possibility for us to achieve conditional decoding. We assume the benign images and unsafe images follow distinctly distinguishable distributions, $\mathbb{X}_n$ and $\mathbb{X}_u$. Thus, the complete encoder feature space $\mathbb{Z}$ can also be divided into benign space and unsafe space, as

$$\mathbb{Z} = \mathbb{Z}_n \cup \mathbb{Z}_u = q_{\phi, x \sim \mathcal{X}_n}\left(z|x\right) \cup q_{\phi, x \sim \mathcal{X}_u}\left(z|x\right). \tag{13}$$

Then we fine-tune the decoder with the loss function like

$$\mathcal{L}(\theta, \phi) = -\mathbb{E}_{z \sim \mathbb{Z}_n}[\log p_\theta\left(x|z\right)] - \mathbb{E}_{z \sim \mathbb{Z}_u}[\log p_\theta\left(\mathbf{0}|z\right)]. \tag{14}$$

The former term guarantees the reconstruction performance for benign features, and the latter term encourages the decoder to decode the unsafe features to zero-vectors. In practice, we describe $\log p_\theta(x|z)$ through the Mean Square Error (MSE). Then the loss of the decoder is crafted as

$$\mathcal{L}(\mathcal{D}_\theta) = \alpha \cdot \frac{1}{|\mathbb{X}_n|} \sum_i \mathcal{L}_{\text{MSE}}\left(\mathcal{D}_\theta\left(\mathcal{E}_\phi\left(x_i\right)\right), x_i\right) + \beta \cdot \frac{1}{|\mathbb{X}_u|} \sum_j \mathcal{L}_{\text{MSE}}\left(\mathcal{D}_\theta\left(\mathcal{E}_\phi\left(x_j\right)\right), \mathbf{0}\right), \tag{15}$$

where $\mathcal{D}_\theta\left(\mathcal{E}_\phi\left(\cdot\right)\right)$ is the encode-decode process. $\mathbb{X}_n$ and $\mathbb{X}_u$ are the defender's training sets of benign and unsafe images. $\alpha, \beta$ controls the weights to combine these two terms.

However, forcing the decoder to map the unsafe features to zeros is strict and superfluous. In fact, we only need to corrupt the decoding outputs from the semantic level, *e.g.*, fuzzy or mosaicked. To this end, we propose the smoothed biased decoding inspired by the VGG perceptual loss in (Johnson

et al., 2016). Specifically, we modify the second term in Equation 15 and get final **conditional decoding loss** as

$$\mathcal{L}_{\mathrm{cd}}(\mathcal{D}_\theta) = \alpha \cdot \frac{1}{|\mathbb{X}_n|} \sum_i \mathcal{L}_{\mathrm{MSE}}\left(\mathcal{D}_\theta\left(\mathcal{E}_\phi\left(x_i\right)\right), x_i\right) + \beta \cdot \frac{1}{|\mathbb{X}_u|} \sum_j \mathcal{L}_{\mathrm{VGG}}\left(\mathcal{D}_\theta\left(\mathcal{E}_\phi\left(x_j\right)\right), \mathbf{0}\right), \quad (16)$$

where $\mathcal{L}_{\mathrm{VGG}}\left(x, \mathbf{0}\right)$ is the smoothed denial-of-service loss that is calculated by

$$\mathcal{L}_{\mathrm{VGG}}\left(x, \mathbf{0}\right) = \mathcal{L}_{\mathrm{MSE}}\left(\mathrm{VGG}\left(x\right), \mathrm{VGG}\left(\mathbf{0}\right)\right), \quad (17)$$

where $\mathrm{VGG}\left(\cdot\right)$ is the feature extractor from the pre-trained VGG-19 model (Simonyan & Zisserman, 2014). We empirically verify that constraining unsafe outputs to be close to zero in the VGG's feature space rather than in the pixel space exhibits less impact on the benign decoding functionality and more stable and generalizable corruption performance of unsafe outputs. Furthermore, since the VGG's feature space aligns with human-perceived attributes (Johnson et al., 2016), ensuring the unlearning effect for unsafe features achieves our intended purpose.

### 4.2.2 FEATURE SPACE CALIBRATION

However, there is a gap between the feature distributions of the encoder output and the diffusion output, leading to PATRONUS's occasional failure in the practical T2I working scenario. We design a feature space calibration to fix the gap, thus generalizing the unlearnability from the encoder feature space to the diffusion feature space. Inspired by the idea of classifier-free guidance (Ho & Salimans, 2022), we introduce text-conditioned features to participate in the conditional decoder's training process.

Specifically, we utilize a caption model $\mathcal{C}$, *e.g.*, LLaVa (Liu et al., 2024), to generate a text description for each image $x_i$ in $\mathbb{X}_u$ and $\mathbb{X}_n$ to serve as the pseudo prompts $p_i$. We input these pseudo prompts into the conditioning module and collect the diffusion outputs to build unsafe and benign diffusion feature set $\mathbb{F}_u$ and $\mathbb{F}_n$ as follows

$$\begin{aligned}
\mathbb{P}\left(\mathcal{X}\right) &= \{p_1, p_2, \ldots, p_n\} = \left\{\mathcal{C}\left(x_i | x_i \in \mathbb{X}\right)\right\}_{i=1,2,\ldots,n}, \\
\mathbb{F}\left(\mathbb{P}, \epsilon_\psi\right) &= \{f_1, f_2, \ldots, f_n\} = \left\{\epsilon_\psi\left(c_i, z_i\right)\right\}_{i=1,2,\ldots,n},
\end{aligned} \quad (18)$$

where $\epsilon_\psi$ is the diffusion module's denoising function parameterized by a U-Net with parameter $\psi$, $c_j$ is the conditional vector obtained by inputting the $j$-th pseudo prompt into the text encoder, $z_j$ is the initial noise.

We compute the **feature-calibration loss** by

$$\mathcal{L}_{\mathrm{fc}} = \frac{1}{|\mathbb{F}_u|} \sum_{f_j \in \mathbb{F}_u} \mathcal{L}_{\mathrm{VGG}}\left(\mathcal{D}_\theta\left(f_j\right), \mathbf{0}\right) + \frac{1}{|\mathbb{F}_n|} \sum_{f_i \in \mathbb{F}_n} \mathcal{L}_{\mathrm{MSE}}\left(\mathcal{D}_0\left(f_i\right), \mathcal{D}_\theta\left(f_i\right)\right), \quad (19)$$

which is combined with the Biased-Decoding loss Equation 16 for fine-tuning the decoder in the image-oriented alignment process. Note that we use the original decoder $\mathcal{D}_0$'s output as the supervision in the second term for two reasons: 1) the original decoder operates well in the diffusion module's feature space; thus it can instruct the feature calibration, and 2) the original decoder has intact benign knowledge, minimizing this loss encourages $\mathcal{D}_\theta$ to continuously review the benign knowledge with the supervision from the teacher $\mathcal{D}_0$.

### 4.3 NON-FINE-TUNABLE SAFETY MECHANISM

As previously discussed, model alignment defenses fail to withstand malicious fine-tuning, and so does our conditional decoder obtained in the last section. To mitigate this vulnerability of alignment models (*e.g.*, the conditional decoder and the aligned U-Net Li et al., 2024; Schramowski et al., 2023; Gandikota et al., 2023), we design an innovative non-fine-tunable safety mechanism, which consists of two parts, *i.e.*, the non-fine-tunability enhancement and the benign performance preservation. The non-fine-tunable safety mechanism combines these two losses and optimizes the model with the objective

$$\mathcal{L}_{\mathrm{nft}} = \gamma \cdot \mathcal{L}_{\mathrm{ftr}} + \lambda \cdot \mathcal{L}_{\mathrm{bpp}}, \quad (20)$$

where $\gamma$ and $\lambda$ are dynamic coefficients computed by an adaptive weights calculator introduced in Appendix D.

### 4.3.1 NON-FINE-TUNABILITY ENHANCEMENT

In this section, we discuss the proposed non-fine-tunability enhancement and its instantiations for the decoder and diffusion modules.

We investigate our goal through the lens of game theory. Take a model (a decoder or a diffusion module) $\mathcal{M}$ as instance, the game between the fine-tuning adversary and the defender can be formulated as a min-max problem like

$$\max_{\mathcal{M}} \mathcal{L} \left( \min_{\phi' \in \Phi'} \mathcal{L}_{\text{MSE}} \left( \phi' \left( \mathcal{M}, \mathbb{X}_m \right), \mathbb{X}_m \right) \right), \tag{21}$$

where $\Phi$ is the fine-tuning strategy set, $\mathbb{X}_m$ is an unsafe data set. In the inner optimization, the adversary tries to fine-tune our model to a state that performs well in the unsafe domain. In the outer optimization, the defender corrupts the states obtained in the inner optimization.

Since we do not assume to know $\mathbb{X}_m$ and $\Phi'$, we leverage the concept of adversarial training to approximate Equation 21. The key to adversarial training is simulating the worst-case adversary in the inner optimization and countering that adversary in the outer optimization. Following this guideline, we craft the following min-max problem to fulfill our goal

$$\max_{\mathcal{M}} \mathcal{L} \left( \min_{\phi \in \Phi} \mathcal{L}_{\text{MSE}} \left( \phi \left( \mathcal{M}, \mathbb{X}_{tune} \right), \mathbb{X}_{eval} \right) \right), \tag{22}$$

where $\mathbb{X}_{tune}$ is the fine-tuning set for inner fine-tuning and $\mathbb{X}_{eval}$ is the evaluation set for outer evaluation. They both come from the defender's unsafe data set $\mathbb{X}_u$, which is sampled from the unsafe domain. $\Phi$ is our simulated fine-tuning strategy set.

Since the max problem in Equation 22 solving is hard to converge, we instead seek to solve the following min-min problem

$$\min_{\mathcal{M}} \mathcal{L}_D \left( \min_{\phi \in \Phi} \mathcal{L}_{\text{MSE}} \left( \phi \left( \mathcal{M}, \mathbb{X}_{tune} \right), \mathbb{X}_{eval} \right), \mathbf{0} \right), \tag{23}$$

where $\mathcal{L}_D$ is a surrogate loss function, which satisfies that minimizing itself shares the similar goal with maximizing the original MSE loss, *i.e.*, disrupting the unsafe outputs. For instance, we can minimize the distance between the model's output and zero vectors, like the biased decoding loss 16 in Section §4.2.

The inner objective represents that the simulated adversary meticulously crafts the fine-tuning strategies to fine-tune our model with the unsafe data $\mathbb{X}_{tune}$. The outer objective represents the defender expects the fine-tuned model to still decode the unsafe images to smoothed zero vectors.

To solve this min-min problem, we utilize the pipeline from (Deng et al., 2024). In practice, we repeat and alternate between inner and outer optimization: At the beginning of each iteration, let $\mathcal{M}_0$ denote the decoder's parameters. First, we use $\mathbb{X}_{tune}$ and strategy $\phi$ to fine-tune $\mathcal{M}_0$ and get the resulting state $\mathcal{M}_1$ as

$$\mathcal{M}_1 = \phi \left( \mathcal{M}_0, \mathbb{X}_{tune} \right). \tag{24}$$

Then, we use $\mathbb{X}_{eval}$ to evaluate $\mathcal{M}_1$'s performance and calculate the **fine-tuning-resistance loss $\mathcal{L}_{\mathbf{r}}$** that measures the discrepancy between the current performance and the desired ones, *e.g.*, outputting zeros when taking the unsafe features as inputs.

Finally, we update $\mathcal{M}_0$ with this loss by doing

$$\theta_0 \leftarrow \theta_0 - \eta \cdot \nabla_{\theta_0} \mathcal{L}_{\text{ftr}}(\mathcal{M}_1, \mathbb{X}_{eval}), \tag{25}$$

where $\theta_0$ is the parameters of $\mathcal{M}_0$ and $\eta$ is the learning rate of the outer optimization.

To save the memory requirements, we turn to first-order approximation (Finn et al., 2017) and update $\mathcal{M}_0$ as follows

$$\theta_0 \leftarrow \theta_0 - \eta \cdot \nabla_{\theta_1} \mathcal{L}_{\text{ftr}}(\mathcal{M}_1, \mathbb{X}_{eval}), \tag{26}$$

That is, we use the gradients with respect to $\mathcal{M}_1$ to approximate the gradients with respect to $\mathcal{M}_0$. Note that the true update to the model is implemented in Equation 26, while the updation between $\mathcal{M}_0$ and $\mathcal{M}_1$ is merely for calculating $\mathcal{L}_r$ and does not modify the model's existent parameters.

Equation 26 makes the model learn the ability to perform poorly when fine-tuned with the unsafe data.

**Mixed Sampling Strategy:** To improve PATRONUS's robustness against different fine-tuning processes, we propose a mixed sampling strategy for the inner loop. To be specific, we construct a bag of fine-tuning strategies containing various optimizers, learning rates, batch sizes, fine-tune sizes, and iteration numbers. Each time, we sample a fine-tuning strategy for the inner loop.

The focus of the bag of fine-tuning strategies is on the selection of the optimizer. To ensure efficiency and effectiveness, we include two optimizers in the inner optimization, *i.e.*, SGD (Robbins & Monro, 1951) and Adam (Kingma & Ba, 2015). These two optimizers have complementary dynamic characteristics, *i.e.*, SGD is better at escaping local optima, while Adam is better at escaping saddle points (Xie et al., 2022). By resisting these two optimizers in the outer optimization, we are able to move our defense model to a state that is difficult to escape and performs poorly on unsafe data.

For other super-parameters like learning rates and batch sizes, we include all commonly used ranges. For fine-tuning size and iteration number, we sample from an excessive range for what is normally required for fine-tuning. We refer to the Appendix G for detailed instantiations of non-fine-tunable decoders and diffusion modules.

### 4.3.2 BENIGN PERFORMANCE PRESERVATION

There is a conflict between the refusal outputs for unsafe inputs and the intact outputs for benign inputs. To preserve the benign performance, we calculate the loss $\mathcal{L}_{\text{bpp}}$ on benign data and it is then combined with the fine-tuning-resistance $\mathcal{L}_{\text{ftr}}$ loss for joint optimization. For the decoder $\mathcal{M}_{dec}$, we compute

$$\mathcal{L}_{\text{bpp}} = \frac{1}{|\mathbb{X}_n|} \sum_{x_i \in \mathbb{X}_n} \mathcal{L}_{\text{MSE}} \left( \mathcal{M}_{dec} \left( \mathcal{E} \left( x_i \right) \right), x_i \right) + \frac{1}{|\mathbb{F}_n|} \sum_{f_i \in \mathbb{F}_n} \mathcal{L}_{\text{MSE}} \left( \mathcal{M}_{dec}^0 \left( f_i \right), \mathcal{M}_{dec} \left( f_i \right) \right),$$

(27)

where the $\mathcal{M}_{dec}^0$ is the original decoder, $\mathcal{E}$ is the corresponding encoder from the VAE. $x_i$ is benign image and $f_i$ is its corresponding diffusion feature (refer to Section §4.2.2). The two terms in Equation 27 encourage the decoder to preserve the benign knowledge in encoder and diffusion feature spaces, respectively.

For the diffusion module, we compute

$$\mathcal{L}_{\text{bpp}} = \frac{1}{|\mathbb{X}_n|} \sum_{x_i \in \mathbb{X}_n} \mathcal{L} \left( \epsilon_\theta \left( \hat{x}_i, c_i, t \right), z \right),$$

(28)

where $\hat{x}_i, c_i, t, z$ are the noisy benign image, conditioning vector from its corresponding caption, timestep, and the ground-truth noise, respectively. Optimizing Equation 28 essentially replicates U-Net's standard training process, which can effectively preserve the benign performance.

## 5 EVALUATION

### 5.1 EXPERIMENT SETUP

**Implementation.** We have implemented a prototype of PATRONUS on the PyTorch (Paszke et al., 2019) platform according to Algorithm 1 using 4 A100-80GB GPUs (NVIDIA). We choose Stable Diffusion (version 1.4) as the experiment subject following previous work (Yang et al., 2023; Gandikota et al., 2023; Li et al., 2024). For the sake of brevity, we refer to the Appendix E for detailed information on PATRONUS' implementation.

**Datasets.** Our experiment involves six datasets, including two image datasets, *i.e.*, ImageNet and NSFW dataset, used in PATRONUS's training process, two sexual-explicit prompt datasets, *i.e.*, I2P and SneakyPrompt, and one sexual-explicit image-caption dataset, *i.e.*, NSFW-prompt, for evaluating the defensive performance of the PATRONUS, and a benign prompt dataset from MS COCO caption dataset for verifying the intactness of benign performance. We refer to the Appendix A for detailed information on these datasets.

**Metrics.** Our experiments involve two metrics that are widely applied in the T2I scenario, *i.e.*,

- **CLIP Score**. The CLIP score assesses the correlation between the image and the corresponding text. It is calculated by the average cosine similarity between the given CLIP text embedding and its generated CLIP image embedding. A higher score is desirable for benign prompts; the opposite is true for unsafe prompts.
- **MSE Error**. For the malicious fine-tuning adversary, we evaluate the fine-tuned model's test loss, illustrating the degree to which the model is optimized in the fine-tuning process. Both the decoder and the diffusion module employ MSE Error as their loss function.

**Baselines.** We compare PATRONUS with five baselines, including SD-V1.4, SD-V2.1, and SLD, and two state-of-the-art model alignment methods, including ESD and SafeGen. We refer to the Appendix C for detailed information on these baselines.

## 5.2 OVERALL PERFORMANCE

In this section, we evaluate the overall performance of PATRONUS compared with baselines. We consider two types of adversaries: 1) the direct prompting adversary who directly prompts the model with unsafe content without modifying its parameters, and 2) the malicious fine-tuning adversary who performs a malicious fine-tuning then prompting.

### 5.2.1 DEFEND AGAINST DIRECT PROMPTING ADVERSARY

This part discusses the defending against the direct prompting adversary. Two representative prompting methods are I2P and SneakyPrompt. Specifically, we employ I2P and SneakyPrompt to query the models and evaluate the CLIP score of the generated images. We present the results in Figure 3 and Figure 4. We can see that PATRONUS achieves the lowest CLIP score compared with the baselines, meaning that PATRONUS generates minimal unsafe content when being maliciously prompted (Figure 3), with even stronger adversarial prompting (Figure 4).

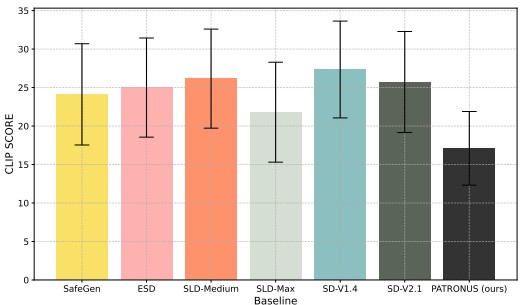 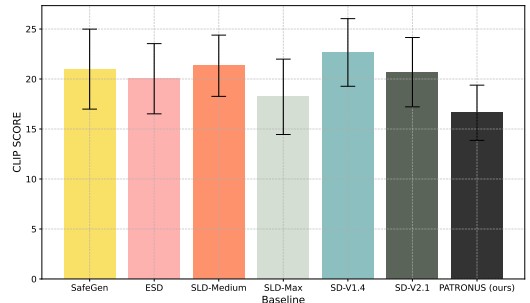

Figure 3: Effectiveness of PATRONUS's defending against I2P attack compared with six baselines. Attacking PATRONUS yields the lowest CLIP score.

Figure 4: Effectiveness of PATRONUS's defending against SneakyPrompt attack compared with six baselines. Attacking PATRONUS yields the lowest CLIP score.

### 5.2.2 DEFEND AGAINST MALICIOUS FINE-TUNING ADVERSARY

Existing work focuses on the threat model like Section §5.2.1, ignoring a realistic but aggressive scenario where the adversary fine-tunes the models with unsafe data before prompting. This section considers the circumstances where the adversary has no knowledge of our defense mechanisms. For instance, in a common scenario, we release our model without sharing defense-related information. Consistent with the usual practice of fine-tuning SD models (*e.g.*, the widely-used diffusers library (von Platen et al., 2022) defaults to fine-tuning the U-Net and does not even provide an interface for fine-tuning other modules), the adversary typically opts to fine-tune the U-Net module. We designate this type of adversary as the naive fine-tuning adversary, and we discuss adaptive fine-tuning adversaries with more prior knowledge in Section H.

To assess the performance of PATRONUS and baselines against this type of adversary, we fine-tune their U-Nets on 200 Porn image-caption pairs from NSFW-prompt for 20 iterations. Then, we use I2P prompt set to query the fine-tuned model and examine the changing trends of the CLIP

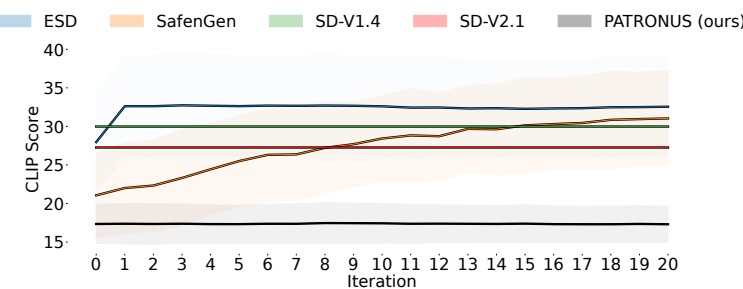

Figure 5: Effectiveness of PATRONUS's resisting malicious fine-tuning compared with four baselines. PATRONUS ensures that the CLIP score on unsafe generations remains consistently low and does not increase as fine-tuning progresses.

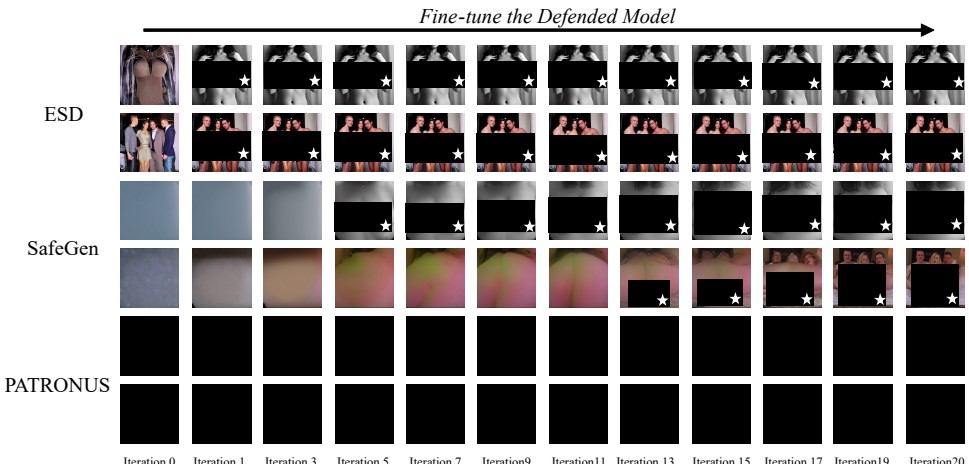

Figure 6: Effectiveness of PATRONUS's resisting malicious fine-tuning compared with four baselines. PATRONUS ensures that the outputs regarding unsafe prompts remain zeros as fine-tuning progresses.

score and the visual results with respect to the number of fine-tuning iterations. A larger CLIP score means more unsafe content in the generated images. From Figure 5, we can see the CLIP scores of SD-V1.4 and SD-V2.1 are high enough from the very beginning, proving the adversarial prompts' effectiveness. The CLIP scores of ESD and SafeGen are low initially, suggesting their effectiveness in defending against adversarial prompts. However, after only a few iterations, their CLIP scores rapidly increase, revealing their vulnerability against malicious fine-tuning. We present the corresponding visual results in Figure 6. As we can see, the CLIP scores of PATRONUS remain low during the whole fine-tuning process, and the generated images are always devoid of unsafe content.

## 5.3 PRESERVING BENIGN PERFORMANCE

In this part, we evaluate the benign performance of benign prompts. Following the existing work, we use captions from MS COCO dataset as benign prompts and examine the CLIP scores. From Figure F, we can see PATRONUS's CLIP scores are comparable to other baselines, demonstrating PATRONUS does not introduce extra degradation on the benign performance.

## 6 CONCLUSION

In this paper, we introduce an innovative defense PATRONUS for pre-trained T2I models, which includes an inseparable moderator and a non-fine-tunable safety mechanism. PATRONUS resolves the drawbacks of existing defenses that fail to remain effective in white-box scenarios. Our experiments have validated the efficacy of PATRONUS in refusing unsafe prompting and resisting malicious fine-tuning as well as its intact benign performance.

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

# A  DATASETS

- **ImageNet**. ImageNet-1k (Deng et al., 2009) is the most commonly used subset of ImageNet, comprising 1000 object classes and 1,281,167 training images, 50,000 validation images, and 100,000 test images. We denote ImageNet as the benign data for the decoder.

- **MS COCO caption dataset**. MS COCO caption dataset (Lin et al., 2014) contains over 330,000 image-caption pairs regarding common objects. We use it to serve as the benign data, participating in the malicious-fine-tuning resistance process of the diffusion. In line with prior works (Li et al., 2024; Schramowski et al., 2023), we build a benign prompt dataset for evaluating the original performance's degradation. We prompt GPT in a template like "You are employing a text-to-image model to generate an image. Describe a scene featuring [object], including details of the background, actions, and expressive adjectives." The [object] is sampled from the categories of ImageNet-1k and MS COCO-2017.

- **NSFW-*porn***. NSFW dataset contains five categories, including *porn, hentai, sexy, normal*. Following the existing work, we focus on the porn class, which has about 50,000 images containing porn semantics. We use NSFW-porn images as the unsafe images for processing the decoder.

- **NSFW-prompt**. SafeGen (Li et al., 2024) creates best prompts for 56k real-world instances of sexual exposure (Bazarov, 2018), based on multiple candidate text captioned by BLIP2 (Li et al., 2023). We adopt a subset of this sexually explicit prompt dataset for the adversary's fine-tuning dataset.

- **I2P**. Inappropriate Image Prompts (AIML-TUDA) are comprised of NSFW text prompts manually tailored on lexica.art. that are deliberately crafted to trick the model into outputting unsafe content. We select all sex-related prompts from this source, resulting in a total of 931 samples. We use this dataset to evaluate the defensive performance.

- **SneakyPrompt**. SneakyPrompt (Yang et al., 2023) utilizes reinforcement learning to generate prompts that can effectively bypass the moderator and manipulate the model's output. As a stronger adversarial attack, this dataset is adopted to evaluate the defensive performance.

Table 1: Effectiveness of PATRONUS against different optimizers.

| Optimizer | Loss in the Unsafe Domain | | | | | |
| | Iteration 0 | Iteration 10 | Iteration 20 | Iteration 30 | Iteration 40 | Iteration 50 |
|---|---|---|---|---|---|---|
| Adade | $0.1267 \pm 1.0$e-4 | $0.1182 \pm 5.5$e-4 | $0.1089 \pm 5.9$e-4 | $0.1010 \pm 9.7$e-4 | $0.0939 \pm 1.2$e-3 | $0.0875 \pm 9.5$e-4 |
| Adam | $0.4093 \pm 6.2$e-3 | $0.0978 \pm 5.72$e-3 | $0.0577 \pm 4.2$e-4 | $0.0503 \pm 2.2$e-3 | $0.0324 \pm 4.3$e-3 | $0.0189 \pm 3.7$e-3 |
| Nes | $0.2388 \pm 6.2$e-2 | $0.3403 \pm 7.3$e-3 | $0.3036 \pm 1.6$e-2 | $0.2315 \pm 8.1$e-2 | $0.1847 \pm 7.5$e-2 | $0.1332 \pm 6.9$e-2 |
| RMS | $0.7481 \pm 2.2$e-1 | $0.0526 \pm 7.0$e-3 | $0.0374 \pm 1.6$e-2 | $0.0250 \pm 5.1$e-3 | $0.0195 \pm 3.0$e-3 | $0.0216 \pm 1.2$e-2 |
| SGD | $0.1807 \pm 4.9$e-2 | $0.3427 \pm 8.6$e-3 | $0.3075 \pm 2.3$e-2 | $0.2541 \pm 5.3$e-2 | $0.2033 \pm 7.7$e-2 | $0.1549 \pm 6.8$e-2 |

Table 2: Effectiveness of PATRONUS against different (potentially) unsafe topics.

| Domain | Loss in the Unsafe Domain | | | | | |
| | Iteration 0 | Iteration 10 | Iteration 20 | Iteration 30 | Iteration 40 | Iteration 50 |
|---|---|---|---|---|---|---|
| NSFW-porn | $0.1807 \pm 4.9$e-2 | $0.3427 \pm 8.6$e-3 | $0.3075 \pm 2.3$e-2 | $0.02541 \pm 5.3$e-2 | $0.2033 \pm 7.7$e-2 | $0.1549 \pm 6.8$e-2 |
| NSFW-sexy | $0.0608 \pm 3.4$e-5 | $0.0518 \pm 3.0$e-4 | $0.0435 \pm 8.9$e-5 | $0.0410 \pm 2.0$e-5 | $0.0398 \pm 7.8$e-5 | $0.0385 \pm 2.8$e-5 |
| Weapon | $0.0333 \pm 1.9$e-6 | $0.0322 \pm 2.8$e-5 | $0.0308 \pm 2.8$e-5 | $0.0298 \pm 1.9$e-5 | $0.0291 \pm 1.9$e-5 | $0.0285 \pm 8.4$e-6 |

Table 3: Effectiveness of PATRONUS against different batch size.

| Batch Size | Loss in the Unsafe Domain | | | | | |
| | Iteration 0 | Iteration 10 | Iteration 20 | Iteration 30 | Iteration 40 | Iteration 50 |
|---|---|---|---|---|---|---|
| 5 | $0.2021 \pm 5.9$e-2 | $0.3409 \pm 1.0$e-2 | $0.3047 \pm 2.0$e-2 | $0.2027 \pm 9.7$e-2 | $0.1662 \pm 8.4$e-2 | $0.1250 \pm 8.1$e-2 |
| 10 | $0.2418 \pm 5.7$e-2 | $0.3216 \pm 4.3$e-2 | $0.3101 \pm 5.3$e-3 | $0.2606 \pm 1.4$e-2 | $0.1813 \pm 6.8$e-2 | $0.1180 \pm 5.8$e-2 |
| 15 | $0.2021 \pm 5.9$e-2 | $0.3409 \pm 1.0$e-2 | $0.3047 \pm 2.0$e-2 | $0.2027 \pm 9.7$e-2 | $0.1662 \pm 8.4$e-2 | $0.1250 \pm 8.1$e-2 |
| 20 | $0.1807 \pm 4.9$e-2 | $0.3427 \pm 8.6$e-3 | $0.3075 \pm 2.3$e-2 | $0.2541 \pm 5.3$e-2 | $0.2033 \pm 7.7$e-2 | $0.1549 \pm 6.8$e-2 |
| 30 | $0.1854 \pm 5.6$e-2 | $0.3400 \pm 7.4$e-3 | $0.3055 \pm 1.9$e-2 | $0.2257 \pm 7.0$e-2 | $0.1923 \pm 6.1$e-2 | $0.1038 \pm 1.2$e-2 |

Table 4: Effectiveness of PATRONUS against different Finetune number.

| Finetune number | Loss in the Unsafe Domain | | | | | |
|---|---|---|---|---|---|---|
| | Iteration 0 | Iteration 10 | Iteration 20 | Iteration 30 | Iteration 40 | Iteration 50 |
| 100 | $0.1873 \pm 4.1e-2$ | $0.3361 \pm 6.3e-3$ | $0.2936 \pm 1.4e-2$ | $0.1934 \pm 7.9e-2$ | $0.1136 \pm 6.2e-2$ | $0.0727 \pm 2.5e-2$ |
| 200 | $0.1599 \pm 3.5e-2$ | $0.3299 \pm 3.4e-3$ | $0.2751 \pm 3.7e-2$ | $0.1424 \pm 8.0e-2$ | $0.0999 \pm 7.8e-2$ | $0.0836 \pm 6.7e-2$ |
| 500 | $0.1807 \pm 4.9e-2$ | $0.3427 \pm 8.6e-3$ | $0.3075 \pm 2.3e-2$ | $0.2541 \pm 5.3e-2$ | $0.2033 \pm 7.7e-2$ | $0.1549 \pm 6.8e-2$ |
| 1000 | $0.2263 \pm 2.5e-2$ | $0.3009 \pm 8.1e-2$ | $0.2619 \pm 1.0e-1$ | $0.2001 \pm 1.0e-1$ | $0.1443 \pm 8.3e-2$ | $0.0959 \pm 7.0e-2$ |
| 2000 | $0.1806 \pm 5.3e-2$ | $0.2742 \pm 9.8e-2$ | $0.2056 \pm 1.3e-1$ | $0.1425 \pm 1.2e-1$ | $0.1211 \pm 9.8e-2$ | $0.0942 \pm 7.7e-2$ |

Table 5: Effectiveness of PATRONUS against different learning rate.

| Learning Rate | Loss in the Unsafe Domain | | | | | |
|---|---|---|---|---|---|---|
| | Iteration 0 | Iteration 10 | Iteration 20 | Iteration 30 | Iteration 40 | Iteration 50 |
| 0.001 | $0.3379 \pm 2.3e-2$ | $0.2074 \pm 3.0e-2$ | $0.0660 \pm 9.7e-3$ | $0.0557 \pm 5.8e-3$ | $0.0500 \pm 2.4e-3$ | $0.0459 \pm 3.0e-3$ |
| 0.00005 | $0.1418 \pm 2.0e-2$ | $0.1835 \pm 8.1e-2$ | $0.1427 \pm 9.4e-2$ | $0.0611 \pm 2.7e-2$ | $0.0463 \pm 5.9e-3$ | $0.0422 \pm 3.1e-3$ |
| 0.0001 | $0.1807 \pm 4.9e-2$ | $0.3427 \pm 8.6e-3$ | $0.3075 \pm 2.3e-2$ | $0.2541 \pm 5.3e-2$ | $0.2033 \pm 7.7e-2$ | $0.1549 \pm 6.8e-2$ |
| 0.002 | $0.3517 \pm 1.2e-2$ | $0.1202 \pm 3.3e-2$ | $0.0681 \pm 1.4e-2$ | $0.0499 \pm 1.5e-3$ | $0.0449 \pm 3.0e-3$ | $0.0394 \pm 2.7e-3$ |
| 0.00001 | $0.1272 \pm 2.4e-4$ | $0.1174 \pm 2.2e-3$ | $0.0974 \pm 2.9e-3$ | $0.0818 \pm 3.3e-3$ | $0.0695 \pm 8.9e-4$ | $0.0616 \pm 3.9e-4$ |

## B  ALGORITHM

---

**Algorithm 1:** PATRONUS

---

**Input:** The benign data $\mathcal{X}_n$ (corresponding benign features $\mathcal{F}_n$), the unsafe data $\mathcal{X}_u$ (corresponding unsafe features $\mathcal{F}_u$), the simulated fine-tuning strategies $\Phi$, the encoder $\mathcal{E}$, MSE loss $\ell$.

**Input:** The pre-trained decoder $\mathcal{D}_0$ and U-Net $\mathcal{U}_0$, the learning rate $\alpha_1$ and iterations $N_1$ for fine-tuning conditional decoder, the learning rate $\alpha_2$ and iterations $N_2$ for NFT enhancement of the decoder and U-Net.

**Output:** The defended decoder $\mathcal{D}$, U-Net $\mathcal{U}$

1 . **Initialize** $\mathcal{D}, \mathcal{U} \leftarrow \mathcal{D}_0, \mathcal{U}_0$.
2 *# Inseparable moderator*
3 **for** 1 to $N_1$ **do**
4     **Sample** a batch of $x_u \sim \mathcal{X}_u$, a batch of $f_u \sim \mathcal{F}_u$, a batch of $x_n \sim \mathcal{X}_n$, a batch of $f_n \sim \mathcal{F}_n$.
5     **Compute**
6     $\mathcal{L}_{\text{cd}} \leftarrow \ell\left(\text{VGG}(\mathcal{D}(\mathcal{E}(x_u))), \text{VGG}(0)\right) + \ell\left(\mathcal{D}(\mathcal{E}(x_n)), x_n\right)$ # biased decoding 4.2.1
7     $\mathcal{L}_{\text{fc}} \leftarrow \ell\left(\text{VGG}(\mathcal{D}(f_u)), \text{VGG}(0)\right) + \ell\left(\mathcal{D}(f_n), \mathcal{D}_0(f_n)\right)$ # feature space calibration 4.2.2
8     $\mathcal{L}_{\text{im}} \leftarrow \alpha \cdot \mathcal{L}_{\text{cd}} + \beta \cdot \mathcal{L}_{\text{fc}}$
9     **Update** $\mathcal{D} \leftarrow \texttt{Adam}(\mathcal{D}, \nabla \mathcal{L}_{\text{im}}, \alpha_1)$
10 **end**
11 *# Non-fine-tunable safety mechanism*
12 **for** $\mathcal{M}$ **in** $[\mathcal{D}, \mathcal{U}]$ **do**
13     **for** 1 to $N_2$ **do**
14         **Sample** one fine-tuning setting $\phi \sim \Phi$
15         **Sample** a batch of $x_{eval} \sim \mathcal{X}_u$, a batch $x_n \sim \mathcal{X}_n$.
16         **for** $k \leftarrow 1$ **to** $K$ **do**
17             # pseudo fine-tuning
18             **Sample** 1 batch of $x_{tune} \sim \mathcal{X}_u$
19             **Fine-tune** $\mathcal{M}_{\vartheta}^k \leftarrow \phi(\mathcal{M}_{\vartheta}^{k-1}, x_{tune})$
20             **Compute**
21             $\mathcal{L}_{i,k} \leftarrow \mathcal{L}_{\text{r}}\left(\mathcal{M}_{\vartheta}^k, x_{eval}\right)$
22         **end**
23         **Compute**
24         $\mathcal{L}_{\text{ftr}} \leftarrow \sum_{k=1}^{K} \mathcal{L}_{i,k}$ # Non-fine-tunability enhancement 4.3.1
25         $\mathcal{L}_{\text{bpp}} \leftarrow \mathcal{L}_{\text{bpp}}(\mathcal{M}, x_n)$ # Benign performance preservation 4.3.2, Note that $\mathcal{M}$ is the statement before entering the pseudo fine-tuning.
26         $\gamma, \lambda \leftarrow \text{MGDA}(\mathcal{L}_{\text{ftr}}, \mathcal{L}_{\text{bpp}})$ # Adaptive weighting D
27         $\mathcal{L}_{\text{nft}} \leftarrow \gamma \cdot \mathcal{L}_{\text{ftr}} + \lambda \cdot \mathcal{L}_{\text{bpp}}$
28         **Update** $\mathcal{M} \leftarrow \texttt{Adam}(\mathcal{M}, \nabla \mathcal{L}_{\text{nft}}, \alpha_2)$
29     **end**
30 **end**

---

## C  BASELINES

- **SD-V1.4** (LMU, b). In accordance with prior research (Li et al., 2024; Gandikota et al., 2023), we utilize the officially supplied Stable Diffusion V1.4.

- **SD-V2.1** (at TU Darmstadt, n.d.). Stable Diffusion 2.1 (SD-V2.1) is retrained on cleansed data, where NSFW information is censored by external safety filters.

- **SLD** (Schramowski et al., 2023). SLD prohibits negative concepts and improves the classifier-free guidance with another diffusion item to shift away from the unsafe domain. We adopt the officially pre-trained model; our configuration examines its four two levels, *i.e.*, medium and max.

- **ESD** (Gandikota et al., 2023). ESD rectifies sexual concepts such as "nudity" to "[blank]" by fine-tuning the cross-attention layers of U-Net. We reproduce ESD train the model for 1000 epochs with a learning rate of $1 \times 10^{-5}$, as the paper suggests.

- **SafeGen** (Li et al., 2024). SafeGen adjusts the diffusion model to corrupt its visual representations related to pornography. We utilize the released model by SafeGen, which has been evaluated in their paper.

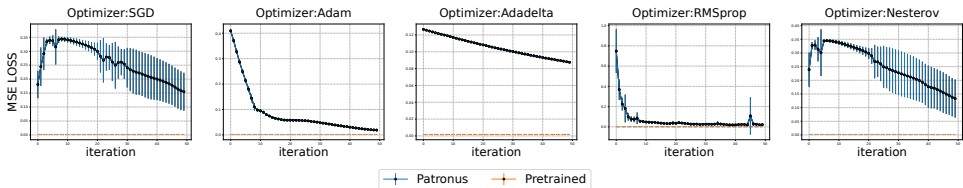

Figure 7: PATRONUS's effectiveness of decoder protection against different optimizers.

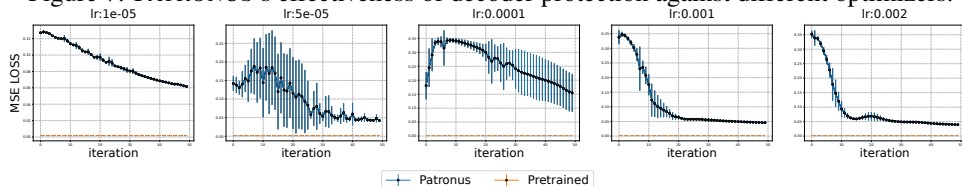

Figure 8: PATRONUS's effectiveness of decoder protection against different learning rates.

## D ADAPTIVE WEIGHTING

In practice, we find it difficult to assign appropriate $\gamma, \lambda$. Therefore, we refer to the Multiple Gradient Descent Algorithm (MGDA), a Multi-task learning technique to optimize a set of (possibly conflicting) objectives (Désidéri, 2012). For tasks $i = 1..k$ with respective losses $\mathcal{L}_i$, it calculates the gradient (separated from the gradients used by the optimizer) for each single task $\nabla \mathcal{L}_i$ and finds the weighting coefficients $\alpha_1..\alpha_k$ that minimize the sum

$$\min_{\alpha_1,...,\alpha_k} \left\{ \left\| \sum_{i=1}^{k} \alpha_i \nabla \mathcal{L}_i \right\|_2^2 \ \middle| \ \sum_{i=1}^{k} \alpha_i = 1, \alpha_i \geq 0 \ \forall i \right\}. \tag{29}$$

In each iteration of non-fine-tunability enhancement, we obtain $\mathcal{L}_{\text{ftr}}$ and $\mathcal{L}_{\text{bpp}}$, then we calculate $\gamma$ and $\lambda$ to strike a balance between $\mathcal{L}_{\text{ftr}}$ and $\mathcal{L}_{\text{bpp}}$, ensuring that the two tasks *i.e.*, the non-fine-tunable enhancement and the benign performance preservation are simultaneously optimized (or at least not degraded).

## E IMPLEMENTATION DETAILS

Given the pre-trained SD model, PATRONUS enhances its decoder and diffusion module and freezes the text encoder. We select the NSFW dataset, especially targeting the porn category, as the unsafe data $\mathbb{X}_u$. We select ImageNet as the benign data $\mathbb{X}_n$ for the decoder, and COCO as the benign data for the diffusion. We adopt LlaVa-13B (Liu et al., 2024) as the caption model to create pseudo prompts. We set the default PATRONUS configuration as $N_1 = 1200$, $N_2 = 1500$, $\alpha_1 = 5e - 5$, $\alpha_2 = 1e - 5$, and $K = 20$. The bag of fine-tuning strategies built for the inner optimization contains the sample options: {Monmentum, Adam} for the optimizer, $\{5 \times 10^{-5}, 10^{-4}, 10^{-3}, 10^{-2}\}$ for the learning rate, $\{4, 8, 12, 16, 20, 24, 30\}$ for the batch size. These options are determined by balancing efficiency and the effectiveness of simulating the adversary.

## F DETAILED PERFORMANCE

## G INSTANTIATIONS OF NON-FINE-TUNABILITY ENHANCEMENT

**Instantiate Non-fine-tunable Decoder:** For the non-fine-tunability enhancement of the decoder $\mathcal{M}_{dec}$, we designate the conditional decoder obtained in Section §4.2 as the starting point. $\mathbb{X}_{tune}$ and $\mathbb{X}_{eval}$ are unsafe image sets. The fine-tuning-resistance loss is calculated by

$$\mathcal{L}_{\text{ftr}} = \sum_{x_i \in \mathbb{X}_{eval}} \mathcal{L}_{\text{VGG}}\left(\mathcal{M}_{dec}\left(x_i\right), \mathbf{0}\right) + \sum_{f_i \in \mathbb{F}_{eval}} \mathcal{L}_{\text{VGG}}\left(\mathcal{M}_{dec}\left(f_i\right), \mathbf{0}\right), \tag{30}$$

$\mathbb{F}_{eval}$ is $\mathbb{X}_{eval}$'s corresponding feature set obtained using the same method described in Section §4.2.2 and used for feature calibration. Optimized with this loss, the decoder learns to decode the unsafe features to smoothed zero vectors after being maliciously fine-tuned.

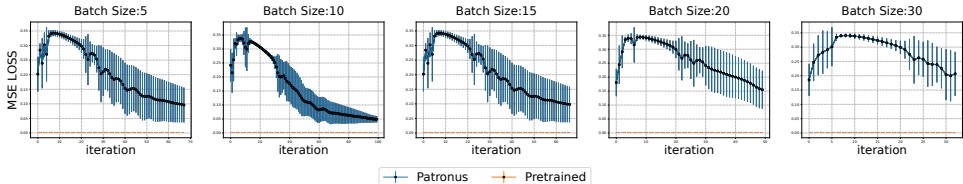

Figure 9: PATRONUS's effectiveness of decoder protection against different batch sizes.

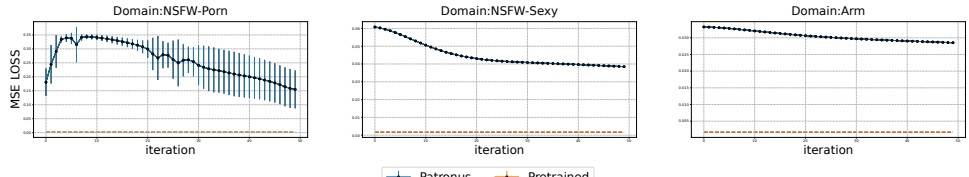

Figure 10: PATRONUS's effectiveness of decoder protection against different unsafe categories.

**Instantiate Non-fine-tunable Diffusion:** For the non-fine-tunability enhancement of the diffusion, we designate our aligned U-Net, which is fine-tuned to consistently predict the noise in unsafe images as zero, as the starting point. $\mathbb{X}_{tune}$ and $\mathbb{X}_{eval}$ are unsafe image-caption sets. Consider the U-Net, parameterized by $\theta$ (noted as $\epsilon_\theta$). $\epsilon_\theta$ predicts the noises added into the images. The fine-tuning-resistance loss is calculated by

$$\mathcal{L}_{\text{ftr}} = \sum_{x_i \in \mathbb{X}_{eval}} \mathcal{L}\left(\epsilon_\theta\left(\hat{x}_i, c_i, t\right), \mathbf{0}\right) \tag{31}$$

where $c_i$ is $x_i$'s corresponding conditioning vector output by the text encoder. Notably, the starting point we chose here is our own aligned model, though theoretically, our non-fine-tunability enhancement method can be compatible with all alignment techniques, such as Li et al. (2024); Schramowski et al. (2023); Gandikota et al. (2023).

## H DEFENDING AGAINST ADAPTIVE ADVERSARY

### H.0.1 MALICIOUS FINE-TUNING TOWARDS CONDITIONAL DECODER

In this part, we assume an adaptive adversary who knows that PATRONUS creates a conditional decoder and utilizes the NSFW-*porn* images to implement more aggressive fine-tuning processes on the decoder. To assess PATRONUS's performance in the worst case, we assume the adversary has already succeeded in compromising the U-Net, leaving only the decoder module to be attacked, *i.e.*, We denote the T2I model with the original U-Net and the conditional decoder as the subject under attack.

We evaluate the robustness of PATRONUS against different fine-tuning strategies, including different optimizers, learning rates, batch sizes, and number of fine-tuning images. We present the MSE losses during the fine-tuning in Figure 7,8,10,11. We can see that PATRONUS introduces significant obstacles to the fine-tuning, making it difficult to converge (resulting in high MSE loss). Simultaneously, it prevents the decoder from generating unsafe content (resulting in always low CLIP scores and corrupted outputs just like Figure 6 shows). Note that PATRONUS shows the robustness against different and unseen fine-tuning settings.

### H.0.2 MALICIOUS FINE-TUNING TOWARDS ALIGNED DIFFUSION

In this part, we assume an adaptive adversary who knows that PATRONUS creates a non-fine-tunable aligned diffusion module and utilizes the NSFW-*prompt* image-caption dataset to implement more aggressive fine-tuning processes on the U-Net. To assess PATRONUS's performance in the worst case, we assume the adversary has already succeeded in compromising the conditional decoder,

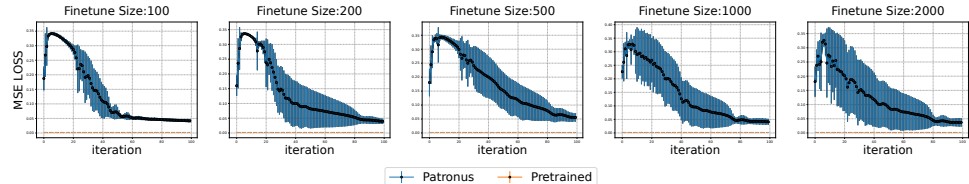

Figure 11: PATRONUS's effectiveness of decoder protection against different fine-tune sizes.

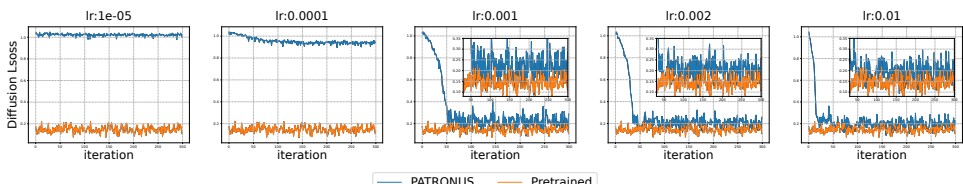

Figure 12: PATRONUS's effectiveness of U-Net protection against different learning rates.

leaving only the U-Net module to be attacked, *i.e.*, We denote the T2I model with the original decoder and the defended U-Net as the subject under attack.

We assume the adversary utilizes 3000 unsafe image-caption pairs to implement the aggressive fine-tuning processes on the U-Net. We evaluate the robustness of PATRONUS against different fine-tuning strategies, including optimizers and learning rates, as shown in 12 and 13. For the learning rates like $1e-5, 1e-4$, PATRONUS leads to the loss remaining nearly unchanged. The bigger learning rates like $0.001, 0.002, 0.01$ allow the loss to drop quickly, they converge at a larger value, leaving the model unable to generate unsafe content. We find it is also the case for RMSprop and Adam optimizers. As for other optimizers, SGD, Adadelta, Nesterov fail to decrease the training loss. We also assess PATRONUS's effectiveness in defending LoRA (Low-Rank Adaptation (Hu et al., 2021)), a popular fine-tuning strategy in the T2I field that introduces two new low-rank parameter matrices for fine-tuning. We test different rank values to validate the robustness of PATRONUS, as shown in Figure 14.

## I    APPLICABILITY FOR VARIOUS UNSAFE CATEGORIES

Given that existing works are often confined to the pornography category, we take a step further and evaluate the application potential of PATRONUS against different unsafe categories. We experiment on NSFW-*sexy* and the weapon dataset (new-workspace bjaa4, 2022) as the image datasets to implement PATRONUS and follow the similar method of I2P to build unsafe prompt sets for evaluating. Here, we consider an adaptive adversary as illustrated in Section §H. We present the adversary's fine-tuning results in Figure 9. As we can see, PATRONUS also showcases the desired rejection of unsafe content and resistance to malicious fine-tuning.

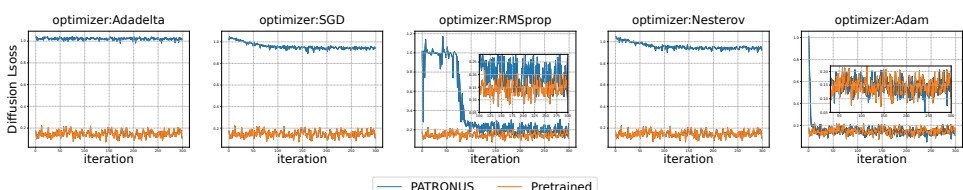

Figure 13: PATRONUS's effectiveness of U-Net protection against different optimizers.

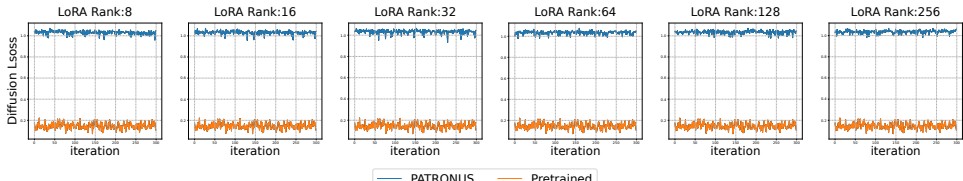

Figure 14: PATRONUS's effectiveness of U-Net protection against different LoRA ranks

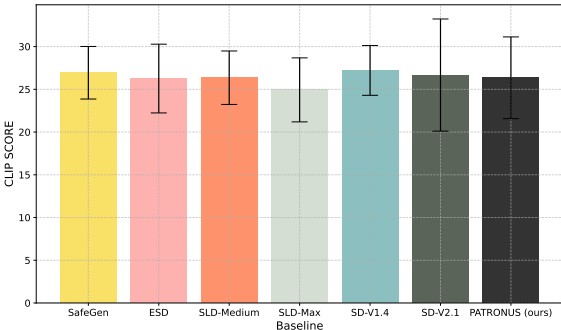

Figure 15: PATRONUS's intact benign performance.

