# OpenReview forum: "Prompt-Independent Safe Decoding to Restrain Unsafe Image Generation for Text-to-Image Models against White-Box Adversary"
_ICLR.cc/2025/Conference — Submitted to ICLR 2025_

### Official Review · Reviewer_RpMp · 2024-10-27

**Soundness:** 4
**Presentation:** 3
**Contribution:** 3
**Rating:** 8
**Confidence:** 5

**Summary:**

This paper presents a framework to protect text-to-image models from generating unsafe content. The framework focuses on three key goals: (1) blocking unsafe content by embedding a safety filter in the decoder, and adjusting the diffusion process to ensure unsafe outputs result in black images, (2) resisting malicious finetuning through adversarial training that simulates such funtuning, and (3) maintaining the quality of benign content by adding a term that preserves it during generation. The design clearly outlines each goal and demonstrates effectiveness through experiments.

**Strengths:**

1. The paper addresses a timely issue—preventing unsafe content generation—and shows effectiveness both in mitigating unsafe outputs and preserving benign ones. It also introduces a defensive measure against adversaries attempting to fintune the model toward unsafe outputs by using an adversarial training scheme.
2. The system is well-designed, with clearly defined objectives, and each loss term is crafted to meet a specific goal, making the approach easy to follow.
3. I appreciate the novel ideas, such as fine-tuning the safety filter within the decoder to act as a conditional decoder and the effective strategy of transforming the min-max optimization into min-min in Equation 23. Both are technically sound and effective solutions.

**Weaknesses:**

1. The non-fine-tunable learning scheme seems largely based on Deng et al. (2024), so I question its contribution as a major novel element. However, I acknowledge that the loss function has been modified for the text-to-image task, providing some level of novelty.

2. The evaluation metrics for defending against unsafe generation only use the CLIP score, but this may not sufficiently distinguish your method from others, given the relatively small range of differences (e.g., values of 17-28 in Figure 3 for seven methods against I2P and 16-23 in Figure 4 for Sneakyprompt). Additional metrics are suggested, like NRR from SafeGen. Also, calculating the similarity between your generated images and a black image (your intended output for unsafe prompts) might provide a more direct assessment.

3. Some strategies lack clear explanations of their benefits and would benefit from experimental support. For example, Section 4.2.2 on Feature Space Calibration and the use of VGG for smoothing could be better justified. Please see detailed questions in the specific sections.

Minor issues:
1. There are inconsistencies between some figures and the descriptions, e.g., L_bd/L_fc in Figure 2 vs L_cd//Lfsc in EQ 11.
2. Some notation in the equations is hard to follow; it would help to provide explanations for each symbol directly near the equation.

I may consider raising my score if my concerns and questions are adequately addressed in the rebuttal.

**Questions:**

1. 4.2.2 (Feature Space Calibration) mentions 'However, there is a gap between the feature distributions of the encoder output and the diffusion output, leading to PATRONUS’s occasional failure in the practical T2I working scenario.' Does this gap occur if the text input is left blank during alignment? Could you clarify the benefit of this calibration—does it primarily enhance unsafe content filtering, benign content preservation, or resistance to malicious fine-tuning? Additionally, could you also explain similar questions on VGG smoothing?

2. The paper mentions that “Φ is our simulated fine-tuning strategy set.” Could you specify the strategies within this set? Also, what impacts might arise if the defender uses different strategies from those of attackers?

---

> ### Author Response · Authors · 2024-11-22
> **Response to Reviewer RpMp**
>
> ## **4.1 Novelty of Non-Fine-Tunability Enhancement**
>
> Yes, Non-Fine-Tunability Enhancement is inspired by Deng et al.’s work. However, existing work focuses on simple classification tasks (10-class datasets) and naïve denoising tasks (on datasets with simple and distinct features like CelebA.), which we find ineffective for T2I tasks:
>
> 1.	It cannot guarantee generalization to a wide range of unsafe semantics. Therefore, we design the smoothed denial-of-service loss for creating the conditional decoder. In this way, we can capture semantic-level features about unsafe generations instead of pixel-level features and then eliminate them.
> 2.	It cannot ensure the robustness against different fine-tuning strategies. Therefore, we propose a mixed sampling strategy for the inner loop. In this way, we expose the model to more diverse fine-tuning strategies in the inner loop, ensuring its defense performance can generalize to a wider range of fine-tuning settings.
> 3.	It cannot preserve the abundant benign semantics. Therefore, we introduce a dynamic weights calculator. We combine the loss from optimizing the Non-Fine-Tunability task and the loss from preserving benign performance and calculate their weights using a dynamic weights calculator. This ensures that both tasks are simultaneously optimized (or at least not degraded).
>
> ## **4.2 Evaluation metric**
>
> We appreciate the reviewer's advice regarding the evaluation metric. After further extensive survey, we have added seven metrics:
>
> •	NRR, Nudity Removal Rate: It refers to the difference in the number of nude parts identified by NudeNet [c1] between the defenses and the SD-V1.4 model.
>
> •	MHSC, Multi-Headed Safety Classifier detection rate
>
> •	ASR
>
> •	TPR
>
> We use these metrics to evaluate the defense performance against malicious prompts.
>
> •	FID
>
> •	LPIPS, Learned Perceptual Image Patch Similarity
>
> •	FPR
>
> We use these metrics to evaluate the benign generations.
> These metrics encompass different defensive preferences (precision and comprehensiveness) and annotators (models and humans). We hope these results address the concerns regarding the metrics. We show the results regarding the I2P attack as follows:
>
> | Baseline | SD-V1.4 | SD-V2.1 | SafeGen | ESD  | SLD-Max | SLD-Medium |   Ours   |
> | :------: | :-----: | :-----: | :-----: | :--: | :-----: | :--------: | :------: |
> |   NRR    |    0    |  0.45   |  0.50   | 0.86 |  0.76   |    0.40    | **0.96** |
> |   MHSC   |  0.38   |  0.21   |  0.35   | 0.06 |  0.09   |    0.24    | **0.02** |
> |   ASR    |  0.40   |  0.32   |  0.40   | 0.09 |  0.11   |    0.33    | **0.03** |
> |   TPR    |  0.65   |  0.72   |  0.70   | 0.90 |  0.86   |    0.69    | **0.99** |
>
> The results regarding the SneakyPrompt attack:
>
> | Baseline | SD-V1.4 | SD-V2.1 | SafeGen |  ESD  | SLD-Max | SLD-Medium |   Ours   |
> | :------: | :-----: | :-----: | :-----: | :---: | :-----: | :--------: | :------: |
> |   NRR    |    0    |  0.79   |  0.83   | 0.93  |  0.79   |    0.21    | **0.99** |
> |   MHSC   |  0.46   |  0.14   |  0.15   | 0.05  |  0.13   |    0.33    | **0.02** |
> |   ASR    |  0.47   |  0.15   |  0.16   | 0.046 |  0.15   |    0.36    | **0.03** |
> |   TPR    |    -    |    -    |  0.88   | 0.95  |  0.87   |    0.65    | **0.98** |
>
> The results regarding the benign performance of COCO dataset:
>
> | Baseline | SD-V1.4 | SD-V2.1 | SafeGen |  ESD  | SLD-Max | SLD-Medium |   Ours    |
> | :------: | :-----: | :-----: | :-----: | :---: | :-----: | :--------: | :-------: |
> |   FID    |  22.4   |  22.5   |  23.6   | 23.7  |  24.1   |    23.4    | **23.6**  |
> |  LPIPS   |  0.78   |  0.77   |  0.78   | 0.79  |  0.81   |    0.79    | **0.78**  |
> |   FPR    |    -    |    -    |  0.010  | 0.012 |  0.023  |   0.018    | **0.011** |
>
> Details:
> •	MHSC detection rate [c2]: MHSC is a binary classifier determining whether the image contains unsafe content.
> •	ASR: We repeat the generation with 5 seeds for each prompt, and the prompt is regarded as a successful attack if at least 1 generation contains unsafe content. We engage four human evaluators to determine the generation's safety.
> •	TPR: We repeat the generation with 5 seeds for each unsafe prompt and calculate the TPR.
> •	FPR: Similarly, we add FPR to evaluate the ratio of benign prompting attempts that unexpectedly trigger the defense mechanism.
>
> [c1] notAI tech. NudeNet: Lightweight Nudity Detection.
>
> [c2] Unsafe Diffusion: On the Generation of Unsafe Images and Hateful Memes From Text-ToImage Models.

---

> ### Author Response · Authors · 2024-11-22
> **Response to Reviewer RpMp**
>
> ## **4.3 Feature space calibration**
>
> Thank you for your inspiring idea that sets the text input null. We haven't tried this idea since the feature space calibration process deals with the decoder module. Nonetheless, it sounds reasonable to process the diffusion module using this idea. Specifically, we can try this: let text input be blank and train the diffusion module to output zero as long as the input images are unsafe. In this way, the diffusion module can be expected to become prompt-independent (i.e., apart from enhancing the diffusion itself, it also qualifies our inseparable decoder's job!)
> Let's return to the feature space calibration process. It has two goals: improve the defensive performance and preserve the benign performance. The intuition is that in case the image-oriented alignment drifts the decoder far away to forget its functionality in the diffusion feature space, we introduce some features from the diffusion to participate in the training to review its knowledge about dealing with the features from the diffusion module, both benign and unsafe features. We have also conducted the related ablation study and will include these results in our revised manuscript if the reviewer suggests doing so.
>
> ## **4.4 VGG smoothed loss**
>
> The perceptual loss (i.e., VGG loss) is from [c4].  To be honest, we turned to perceptual loss and found it work well without coming up with any additional adjustments after encountering failure with the mere MSE loss. We do consider "why it works". The perceptual loss is devised to overcome the shortcoming that the per-pixel losses often miss the perceptual relationship between images, which fits our needs, as it helps eliminate unsafe content at the semantic feature level based on what is visually perceived by the human. We also have the related ablation study results and will include them in our revised manuscript if the reviewer advises us to do so.
>
> [c4] Perceptual losses for real-time style transfer and super-resolution
>
> ## **4.5 Simulated fine-tuning strategy set**
>
> We apologize for the excessive abbreviation of this part; we will move the related content from the appendix to the main body.
> The simulated fine-tuning strategy set contains three dimensions: optimizer, learning rate, and batch size. Each dimension has an optional range: the optimizer can be SGD or Adam; the learning rate is sampled from [5e-5, 1e-2]; the batch size is sampled from [4, 32]. The main goal is to cover as many fine-tuning strategies as possible that an actual attacker might choose in the future.
> Nonetheless, we cannot expect to exhaust all fine-tuning possibilities, so we need to ensure the generalization. Specifically, we cover the commonly used ranges in our fine-tuning strategy set for hyperparameters like learning rate and batch size. As for the optimizer, we verify that alternating between SGD and Adam is sufficient to ensure good generalization. This success is due to the complementary dynamics of the two optimizers [c5]. By using both in the inner loop, we can move the model to a local optimum that is harder to escape from [c6], i.e., achieving generalizable Non-Fine-Tunable learning.
>
> [c5] Adaptive inertia: Disentangling the effects of adaptive learning rate and momentum. In International conference on machine learning
>
> [c6] SOPHON: Non-Fine-Tunable Learning to Restrain Task Transferability For Pre-trained Models
>
> ## **4.6 Miscellany**
>
> We sincerely thank the reviewer for the suggestions. We will fix the mentioned figure-equation mismatch issue and check all the others. We will adjust the occurrence of the symbol explanations to improve the readability. We will also conduct a thorough review to identify any issues that might cause difficulties for readers.

---

> > ### Comment · Reviewer_RpMp · 2024-11-24
> >
> > Thanks for your reply. My questions have been addressed, so I will increase my rating.

---

> > > ### Author Response · Authors · 2024-11-28
> > > **Thanks for your support**
> > >
> > > Thanks for your prompt response despite such a busy period. We deeply appreciate your support and we will try our best to keep improving our work. Have a good day~

---

### Official Review · Reviewer_BgyQ · 2024-11-01

**Soundness:** 3
**Presentation:** 2
**Contribution:** 3
**Rating:** 6
**Confidence:** 3

**Summary:**

The paper proposes PATRONUS, a new defense framework that protects text-to-image models from white-box adversarial attacks. With a special emphasis on the threats resulting from model fine-tuning, PATRONUS represents an internal moderator and a mechanism of non fine-tunable learning rather than traditional defenses such as content moderation or alignment, which could be manipulated by attackers even at the parameter level. The internal moderator transforms unsafe input features into zero vectors, ensuring that benign features are decoded correctly while rejecting unsafe prompts. In this regard, the authors prove the efficacy of PATRONUS against different adversarial attacks with much stronger robustness compared to the state-of-the-art methods by retaining low CLIP scores on unsafe inputs while keeping performance on benign inputs.

**Strengths:**

The paper's main strengths lie in its method of applying the concepts of non-fine-tunable learning and inseparable content moderation to white-box adversaries in T2I models. it demonstrates a clear understanding of model architecture, with the authors skillfully leveraging each component to optimize the defense mechanism effectively.

**Weaknesses:**

One limitation to the method could be expensive computational resources. Using 4 A100-80GB GPUs might not be ideal for all the users who wish to use this defense method.

**Questions:**

Would the performance of non-fine-tunable mechanism maintain its effectiveness if PATRONUS is implemented on smaller or resource-limited T2I models?

---

> ### Author Response · Authors · 2024-11-22
> **Response to Reviewer BgyQ**
>
> ## **3.1 Implementation cost**
>
> Sorry for causing a misunderstanding. We use 4 A100-80G GPUs in our experiments but the minimum requirements of performing our defense are 4\*24G (for processing the decoder) and 4\*10G (for processing the diffusion module). The cost of our method is close to training a T2I model since we reduce computational demands by introducing LoRA for processing the diffusion module and turning to first-order approximation when performing the Non-Fine-Tunable learning's bi-level optimization.
>
> Since the LoRA rank we used in the experiment is only 8, this means that we only need a small parameter space to achieve Non-Fine-Tunability. Therefore, our method holds great potential for successful implementation on smaller or resource-limited models.

---

### Official Review · Reviewer_k1Hz · 2024-11-01

**Soundness:** 2
**Presentation:** 3
**Contribution:** 2
**Rating:** 3
**Confidence:** 4

**Summary:**

This paper tries to propose a method to align text-to-image model to prevent unsafe image generation. The idea is to fine-tune the decoder and diffusion model (U-Net). Key challenge for alignment is that if the model is open-source, an attacker may fine-tune the model to remove the alignment. This paper aims to solve this challenge via aligning the model in a specific way. The alignment itself is a fine-tuning process. Evaluation is performed on several datasets and some malicious fine-tuning.

**Strengths:**

This paper aims to solve a challenging problem. For an open-source model, malicious fine-tuning to remove the safety alignment is a key challenge.

The paper follows a good formulation.

Evaluation is conducted on multiple datasets, and multiple baseline alignment methods are compared.

**Weaknesses:**

The most important weakness is that the reviewer is not convinced the proposed method solves the challenge. It is not clear why the proposed alignment method can survive malicious fine-tuning. It's not clear to why the proposed method is different from baselines in terms of resistance against malicious fine-tuning.

Evaluation metrics are too simple. Mainly CLIP is used to assess both attack effectiveness and performance preservation. More metrics are expected, e.g., the number of nudity parts in a generated image. FID scores or LPIPS scores to measure performance preservation for benign prompts.

Adaptive attacks are shown in Appendix. I would say these are the most important results for such a defense work. I suggest to focus on these adaptive attacks, and show detailed results in main body using more fine-grained metrics for both attack effectiveness and performance preservation.

Eventually, a strong attacker can remove any safety alignment with enough fine-tuning data. An extreme case is that the strong attacker re-trains a model without safety alignment given enough tuning data and computation resources. So, a right question to ask is how much more resources (data and computation) an attacker needs in order to remove the safety alignment. The paper can benefit from showing such experimental results.

It's also helpful to show images generated for unsafe prompts (adaptive attacks) and safe prompts, for different methods.

**Questions:**

Please see the above weaknesses.

---

> ### Author Response · Authors · 2024-11-22
> **Response to Reviewer k1Hz**
>
> ## **2.1 How the Non-Fine-Tunable learning works**
>
> We sincerely thank the reviewer for pointing out this potential confusion. The intuition of our non-fine-tunable protection mechanism aligns with the previous work [c1][c2] that mainly focus on image classification. The main idea is to simulate adversary’s malicious fine-tuning in the inner loop, and corrupt the outcome performance of fine-tuning in the outer loop, which is inspired by and developed from model agnostic meta-learning (MAML). We will incorporate more preliminary information (including intuition) about Non-Fine-Tunability Enhancement in our revised manuscripts.
>
> [c1] SOPHON: Non-Fine-Tunable Learning to Restrain Task Transferability For Pre-trained Models
>
> [c2] Self-destructing models: Increasing the costs of harmful dual uses of foundation models
>
> ## **2.2 Evaluation metric**
>
> We appreciate the reviewer's advice regarding the evaluation metric. After further extensive survey, we have added seven metrics:
>
> •	NRR, Nudity Removal Rate: It refers to the difference in the number of nude parts identified by NudeNet [c1] between the defenses and the SD-V1.4 model.
>
> •	MHSC, Multi-Headed Safety Classifier detection rate
>
> •	ASR
>
> •	TPR
>
> We use these metrics to evaluate the defense performance against malicious prompts.
>
> •	FID
>
> •	LPIPS, Learned Perceptual Image Patch Similarity
>
> •	FPR
>
> We use these metrics to evaluate the benign generations.
> These metrics encompass different defensive preferences (precision and comprehensiveness) and annotators (models and humans). We hope these results address the concerns regarding the metrics. We show the results regarding the I2P attack as follows:
>
> | Baseline | SD-V1.4 | SD-V2.1 | SafeGen | ESD  | SLD-Max | SLD-Medium |   Ours   |
> | :------: | :-----: | :-----: | :-----: | :--: | :-----: | :--------: | :------: |
> |   NRR    |    0    |  0.45   |  0.50   | 0.86 |  0.76   |    0.40    | **0.96** |
> |   MHSC   |  0.38   |  0.21   |  0.35   | 0.06 |  0.09   |    0.24    | **0.02** |
> |   ASR    |  0.40   |  0.32   |  0.40   | 0.09 |  0.11   |    0.33    | **0.03** |
> |   TPR    |  0.65   |  0.72   |  0.70   | 0.90 |  0.86   |    0.69    | **0.99** |
>
> The results regarding the SneakyPrompt attack:
>
> | Baseline | SD-V1.4 | SD-V2.1 | SafeGen |  ESD  | SLD-Max | SLD-Medium |   Ours   |
> | :------: | :-----: | :-----: | :-----: | :---: | :-----: | :--------: | :------: |
> |   NRR    |    0    |  0.79   |  0.83   | 0.93  |  0.79   |    0.21    | **0.99** |
> |   MHSC   |  0.46   |  0.14   |  0.15   | 0.05  |  0.13   |    0.33    | **0.02** |
> |   ASR    |  0.47   |  0.15   |  0.16   | 0.046 |  0.15   |    0.36    | **0.03** |
> |   TPR    |    -    |    -    |  0.88   | 0.95  |  0.87   |    0.65    | **0.98** |
>
> The results regarding the benign performance of COCO dataset:
>
> | Baseline | SD-V1.4 | SD-V2.1 | SafeGen |  ESD  | SLD-Max | SLD-Medium |   Ours    |
> | :------: | :-----: | :-----: | :-----: | :---: | :-----: | :--------: | :-------: |
> |   FID    |  22.4   |  22.5   |  23.6   | 23.7  |  24.1   |    23.4    | **23.6**  |
> |  LPIPS   |  0.78   |  0.77   |  0.78   | 0.79  |  0.81   |    0.79    | **0.78**  |
> |   FPR    |    -    |    -    |  0.010  | 0.012 |  0.023  |   0.018    | **0.011** |
>
> Details:
> •	MHSC detection rate [c2]: MHSC is a binary classifier determining whether the image contains unsafe content.
> •	ASR: We repeat the generation with 5 seeds for each prompt, and the prompt is regarded as a successful attack if at least 1 generation contains unsafe content. We engage four human evaluators to determine the generation's safety.
> •	TPR: We repeat the generation with 5 seeds for each unsafe prompt and calculate the TPR.
> •	FPR: Similarly, we add FPR to evaluate the ratio of benign prompting attempts that unexpectedly trigger the defense mechanism.
>
> [c1] notAI tech. NudeNet: Lightweight Nudity Detection.
>
> [c2] Unsafe Diffusion: On the Generation of Unsafe Images and Hateful Memes From Text-ToImage Models.
>
> ## **2.3 Adaptive attacks**
>
> Due to page constraints, we placed the evaluation against the adaptive attacker in the appendix. We will move this part to the main body in the revised manuscript to provide a more detailed analysis.
>
> ## **2.4 Attack Cost**
>
> We thank the reviewer for reminding us to provide a clearer attack cost. Based on our experiment, attackers with fewer resources than 2000 fine-tuning samples and 500 fine-tuning iterations, cannot corrupt our defense. And the defensive performance begins to faint when the resource outruns this budget. In comparison, existing works begin to faint with only 200 samples and less than 20 iterations.
> We will incorporate more detailed analysis, including the lower bound and the “Attack resource-Defense performance” curve, which we believe can provide more constructive guidance for relevant practitioners.
>
> ## **2.5 Show some benign generations**
>
> Yeah, sure. We omitted this part due to space limitations. We will include some benign generations in our revised manuscript.

---

> > ### Comment · Reviewer_k1Hz · 2024-11-26
> > **Thanks for the responses**
> >
> > Thank you for the response. The preliminary experimental results on additional evaluation metrics is appreciated, and these metrics provide helpful insights. I believe incorporating them consistently across all evaluations could enhance the paper. For instance, although the paper's method achieves comparable LPIPS with baselines, LPIPS still seems quite large. However, the more critical issues highlighted in my initial review remain unaddressed. As a result, I will maintain my original score.

---

> > > ### Author Response · Authors · 2024-11-26
> > > **Thanks for the feedback**
> > >
> > > Thank you for pointing out your concerns further! Here, we provide further explanations regarding sections 2.1 and 2.4 of our last response.
> > >
> > > ## **2.1 How it works**
> > >
> > > To clearly state the intuition of our method, we first review the intuition of GAN. GAN formulates a min-max game where the discriminator discriminates between real and artificial outputs ("max optimization"), and the generator is trained to defeat the discriminator adversarially ("min optimization"). Similarly, the main idea of our method is to simulate the adversary's malicious fine-tuning in the inner "max optimization" and corrupt the outcome performance of fine-tuning in the outer "min optimization" adversarially.
> > >
> > > When it comes to other baselines, such as ESD and SafeGen, their defensive knowledge can be easily corrupted by fine-tuning because their models have not been preemptively trained to identify and defend such malicious fine-tuning processes. Let's put it this way: although the models from these baselines initially exhibit defensive behaviors (i.e., refusing to generate unsafe content), they treat the malicious fine-tuning by the adversary as just a normal fine-tuning process (e.g., a benign fine-tuning with Pokemon images). As a result, they will be easily fine-tuned to the new, unwanted downstream tasks, i.e., generating unsafe content. In contrast, our method produces a model that not only rejects unsafe generation under normal conditions but also detects malicious fine-tuning attempts as harmful, thus resisting them. This makes it much harder to fine-tune the model to perform unsafe generation, as if it has been located at a local optimum on the surface of the unsafe task.
> > >
> > > We hope the analogies could address your concern.
> > >
> > >
> > >
> > > ## **2.4 Attack cost**
> > >
> > > Sorry for the unclear presentation about this part. In the initial submission, we showed the experimental results regarding how much resources (data and computation) an attacker needs in order to remove our safety alignment in Figure 11 in the appendix. Based on our further experiment, attackers with fewer resources than 2000 fine-tuning samples and 500 fine-tuning iterations (shown in Table X below), cannot corrupt our defense. In comparison, existing works begin to faint with only 200 samples and less than 20 iterations, as shown in Figure 6. It empirically shows our method increases the attack cost for potential adversaries by 10x (data) and 25x (computation).  We will move these results to the main body and provide a detailed analysis in our revised manuscript.
> > >
> > > Table X: Training loss over iterations
> > >
> > > | Baseline\Iteration | 0              | 50             | 100             | 150            | 200             | 250             | 300             | 350             | 400             | 450             | 500             |
> > > | ------------------ | -------------- | -------------- | --------------- | -------------- | --------------- | --------------- | --------------- | --------------- | --------------- | --------------- | --------------- |
> > > | Undefended         | 0.0017$\pm$0.0 | -              | -               | -              | -               | -               | -               | -               | -               | -               | -               |
> > > | Ours               | 0.18$\pm$0.049 | 0.15$\pm$0.065 | 0.055$\pm$0.014 | 0.042$\pm$0.01 | 0.036$\pm$0.011 | 0.030$\pm$0.012 | 0.025$\pm$0.012 | 0.021$\pm$0.012 | 0.018$\pm$0.011 | 0.015$\pm$0.010 | 0.014$\pm$0.010 |
> > >
> > > *note: the usable unsafe generation requires an MSE loss of less than 0.005.
> > >
> > >
> > >
> > > We hope our explanation addresses your concerns satisfactorily. If you have further questions, please let us know~

---

> > > > ### Author Response · Authors · 2024-12-01
> > > > **A kind reminder for approaching discussion deadline**
> > > >
> > > > Dear Reviewer k1Hz,
> > > >
> > > > We sincerely appreciate your valuable time and effort in reviewing our manuscript and offering constructive comments, according to which we have tried our best to improve our work.
> > > >
> > > > As the author-reviewer discussion phase is drawing to a close, we would like to confirm whether our responses have effectively addressed your concerns. Here is a summary of our response for your convenience:
> > > >
> > > > 1. **Working mechanism:** We provided a more detailed explanation about the intuition of our method and compared our method with other alignment-based baselines, answering why our method can defend against the white-box adversary.
> > > > 2. **Evaluation metric:** We have added seven metrics as you recommended, including NRR, MHSC, ASR, TPR, FID, LPIPS, and FPR, to verify the defense performance and benign performance.
> > > > 3. **Adaptive attacks:** We will move this part to the main body and provide a more detailed analysis.
> > > > 4. **Attack cost:** We provided further experimental results about the resources (data and computation) that an adversary needs to corrupt our model.
> > > >
> > > > We sincerely hope our responses can address your concern. If you require further clarification or have any additional concerns, please do not hesitate to contact us. We are more than willing to continue our communication with you.
> > > >
> > > > I hope my response hasn’t disturbed you. Have a great weekend！
> > > >
> > > > Best regards,
> > > >
> > > > Authors of #14032

---

### Official Review · Reviewer_My5p · 2024-11-03

**Soundness:** 2
**Presentation:** 3
**Contribution:** 3
**Rating:** 3
**Confidence:** 4

**Summary:**

This paper introduces a novel defensive solution called Patronus, designed to prevent T2Is from generating unsafe content. Patronus integrates an output moderator directly within the T2I's decoder. It guides the image decoder to perform conditional decoding based on safety features, transforming unsafe input features into zero vectors.

**Strengths:**

1. This paper addresses a crucial safety issue: defending T2I models against white-box adversaries. As T2I models become more popular and widely deployed, this topic is increasingly important.

2. The paper presents a novel and intriguing idea.

3. The content is well-structured and easy to follow.

**Weaknesses:**

While the paper introduces 'Patronus' as a novel defensive mechanism, the evaluation section of this paper is too weak, making the effectiveness of the proposed method unclear, which is essential for achieving a higher-scoring paper.

1. Lacking Evaluation Results on SOTA adversarial attack for T2I. The authors should assess the proposed defense against leading Text-to-Image adversarial attacks, such as MMA-Diffusion [c0]. This attack has been reported to outperform I2P, posing a significant threat to T2I models. Evaluating the defense's effectiveness by reporting the Attack Success Rate after implementing Patronus would demonstrate its generalizability and robustness.

[c0] https://arxiv.org/abs/2311.17516

2. The related work section overlooks a key contribution, [c1], which fine-tunes a text decoder to defend against adversarial prompts. In contrast to this approach, the present work focuses on fine-tuning the image decoder. A discussion of [c1] should be included in the related work, highlighting this crucial difference in methodology and its potential implications. This comparison will provide a more complete context for the current work and clarify its unique contribution.

[c1] https://arxiv.org/pdf/2403.01446

3. Lacking comparison with close-related baselines.  The authors assert that existing content moderators "can be easily removed by white-box adversaries" and have other drawbacks (lines 037-041). However, it's unclear how effectively the proposed "Patronus" addresses these issues. To demonstrate its superiority, it is recommended that the authors compare their method with existing content moderation solutions such as LlamaGuard [c2] and OpenAI-Moderation [c3]. This comparison would highlight the effectiveness of Patronus in overcoming these challenges.

[c2] https://arxiv.org/abs/2312.06674

[c3] https://arxiv.org/pdf/2208.03274

4. Limited Evaluation Metrics: Relying solely on "CLIPScore" offers an incomplete view of the proposed solution's performance. The error bar in Figure 3 nearly obscures the performance gain. A more comprehensive evaluation should include metrics such as Attack Success Rate (ASR), Area Under the Receiver Operating Characteristic Curve (AUROC), and False Positive Rate (FPR), as recommended in [c1], [c2], and [c3]. These metrics provide a clearer understanding of the solution's impact on both normal and unsafe use cases.

5. Absence of Adaptive Attack Evaluation: The evaluation doesn't consider the critical scenario of adaptive attacks, where the attacker possesses full knowledge of both the T2I model and the implemented safeguards. Evaluating the defense's resilience against adaptive attacks is crucial for understanding its real-world effectiveness. The authors can follow the adaptive attack design principles suggested in [c4] to conduct experiments.

[c4] Tramer, F., Carlini, N., Brendel, W., & Madry, A. (2020). On adaptive attacks to adversarial example defenses. Advances in neural information processing systems, 33, 1633-1645.

**Questions:**

Please refer to the weakness section.

---

> ### Author Response · Authors · 2024-11-22
> **Response to Reviewer My5p**
>
> ## **1.1 SOTA attacks**
>
> We appreciate the suggestion about evaluating the defense against MMA-diffusion. The comparison results between our method and other baselines:
>
> | Baseline | SD-V1.4 | SD-V2.1 | SafeGen |  ESD  | SLD-Max | SLD-Medium | Ours  |
> | :------: | :-----: | :-----: | :-----: | :---: | :-----: | :--------: | :-------: |
> |   NRR    |    0    |  0.85  |  0.96  | 0.97 |  0.72  |   0.18    | **1.0** |
> |   MHSC   |  0.84  |  0.18  |  0.06  | 0.17 |  0.36  |   0.76    | **0.02** |
> |   ASR    |  0.85  |  0.21  |  0.10  | 0.28 |  0.41  |   0.79    | **0.01** |
> |   TPR    |    -    |    -    |  0.82  | 0.92 |  0.74  |   0.69    | **0.99** |
>
> In the revised manuscript, we will include these results in the evaluation section.
>
> ## **1.2 Related work**
>
> Thank you for reminding us of this relevant work. Yang et al. utilize a LLM, which decodes the text guidance embeddings to natural language, to safeguard the T2I. It belongs to the moderator-based family since the embedding decoder is an external module. The difference between our defense and moderator-based defenses is presented in the next section. We will include this work and its discussion in our revised manuscript.
>
> ## **1.3 Moderator-based baselines**
>
> In our scenario, the adversary can directly remove any external moderator with white-box access, e.g., by commenting out the relevant code. Since moderator-based methods will fail in this scenario, we did not use them as our baselines. In contrast, Patronus is intrinsic and inseparable in the model parameters and thus cannot be removed directly. We evaluated Patronus by comparing it with SOTA alignment-based methods, validating its effectiveness and robustness.
>
> ## **1.4 Evaluation metric**
>
> We appreciate the reviewer's advice regarding the evaluation metric. After further extensive survey, we have added seven metrics:
>
> •	NRR, Nudity Removal Rate: It refers to the difference in the number of nude parts identified by NudeNet [c1] between the defenses and the SD-V1.4 model.
>
> •	MHSC, Multi-Headed Safety Classifier detection rate
>
> •	ASR
>
> •	TPR
>
> We use these metrics to evaluate the defense performance against malicious prompts.
>
> •	FID
>
> •	LPIPS, Learned Perceptual Image Patch Similarity
>
> •	FPR
>
> We use these metrics to evaluate the benign generations.
> These metrics encompass different defensive preferences (precision and comprehensiveness) and annotators (models and humans). We hope these results address the concerns regarding the metrics. We show the results regarding the I2P attack as follows:
>
> | Baseline | SD-V1.4 | SD-V2.1 | SafeGen |  ESD  | SLD-Max | SLD-Medium | Ours  |
> | :------: | :-----: | :-----: | :-----: | :---: | :-----: | :--------: | :-------: |
> |   NRR    |    0    |  0.45  |  0.50  | 0.86 |  0.76  |   0.40    | **0.96** |
> |   MHSC   |  0.38  |  0.21  |  0.35  | 0.06 |  0.09  |   0.24    | **0.02** |
> |   ASR    |  0.40  |  0.32  |  0.40  | 0.09 |  0.11  |   0.33    | **0.03** |
> |   TPR    |  0.65  |  0.72  |  0.70  | 0.90 |  0.86  |   0.69    | **0.99** |
>
> The results regarding the SneakyPrompt attack:
>
> | Baseline | SD-V1.4 | SD-V2.1 | SafeGen |  ESD   | SLD-Max | SLD-Medium | Ours  |
> | :------: | :-----: | :-----: | :-----: | :----: | :-----: | :--------: | :-------: |
> |   NRR    |    0    |  0.79  |  0.83  | 0.93  |  0.79  |   0.21    | **0.99** |
> |   MHSC   |  0.46  |  0.14  |  0.15  | 0.05  |  0.13  |   0.33    | **0.02** |
> |   ASR    |  0.47  |  0.15  |  0.16  | 0.046 |  0.15  |   0.36    | **0.03** |
> |   TPR    |    -    |    -    |  0.88  | 0.95  |  0.87  |   0.65    | **0.98** |
>
> The results regarding the benign performance of COCO dataset:
>
> | Baseline | SD-V1.4 | SD-V2.1 | SafeGen |  ESD  | SLD-Max | SLD-Medium | Ours  |
> | :------: | :-----: | :-----: | :-----: | :---: | :-----: | :--------: | :-------: |
> |   FID    |  22.4  |  22.5  |  23.6  | 23.7 |  24.1  |   23.4    | **23.6** |
> |  LPIPS   |  0.78  |  0.77  |  0.78  | 0.79 |  0.81  |   0.79    | **0.78** |
> |   FPR    |    -    |    -    |  0.010  | 0.012 |  0.023  |   0.018    | **0.011** |
>
> Details:
> •	MHSC detection rate [c2]: MHSC is a binary classifier determining whether the image contains unsafe content.
> •	ASR: We repeat the generation with 5 seeds for each prompt, and the prompt is regarded as a successful attack if at least 1 generation contains unsafe content. We engage four human evaluators to determine the generation's safety.
> •	TPR: We repeat the generation with 5 seeds for each unsafe prompt and calculate the TPR.
> •	FPR: Similarly, we add FPR to evaluate the ratio of benign prompting attempts that unexpectedly trigger the defense mechanism.
>
> [c1] notAI tech. NudeNet: Lightweight Nudity Detection.
>
> [c2] Unsafe Diffusion: On the Generation of Unsafe Images and Hateful Memes From Text-ToImage Models.
>
> ## **1.5 Adaptive attacks**
>
> Due to page constraints, we placed the evaluation against the adaptive attacker in the appendix. We will move this part to the main body and provide a more detailed analysis.

---

> > ### Author Response · Authors · 2024-12-01
> > **A kind reminder for approaching discussion deadline**
> >
> > Dear Reviewer My5p,
> >
> > We sincerely appreciate your valuable time and effort in reviewing our manuscript and offering constructive comments, according to which we have tried our best to improve our work.
> >
> > As the author-reviewer discussion phase is drawing to a close, we would like to confirm whether our responses have effectively addressed your concerns. Here is a summary of our response for your convenience:
> >
> > 1. **SOTA attack:** We conducted additional experiments regarding MMA-diffusion.
> > 2. **Related work and moderator-based baselines:**  We provided a more detailed explanation about our working scenario, i.e., defending white-box adversary, by comparing our defense with the close-related moderator-based baselines and a related work, LLM decoder by Yang et al.
> > 3. **Evaluation metric:** We have added seven metrics as you recommended, including NRR, MHSC, ASR, TPR, FID, LPIPS, and FPR, to verify the defense performance and benign performance.
> > 4. **Adaptive attacks:** We placed the evaluation against the adaptive attacker in the appendix. We will move this part to the main body and provide a more detailed analysis.
> >
> > We sincerely hope our responses can address your concern. If you require further clarification or have any additional concerns, please do not hesitate to contact us. We are more than willing to continue our communication with you.
> >
> > I hope my response hasn’t disturbed you. Have a great weekend！
> >
> > Best regards,
> >
> > Authors of #14032

---

### Author Response · Authors · 2024-11-22
**Overall Response**

Dear Reviewers, Area Chairs, and Program Chairs,

We sincerely thank all four reviewers for their constructive comments and insightful questions, which helped us refine our work. Reviewers have acknowledged the impact and novelty of our work and the comprehensive analysis.

**[Problem Importance]**:

+ **Reviewer My5p**: This paper addresses a crucial safety issue: defending T2I models against white-box adversaries. As T2I models become more popular and widely deployed, this topic is increasingly important.
+ **Reviewer k1Hz**: This paper aims to solve a challenging problem. For an open-source model, malicious fine-tuning to remove the safety alignment is a key challenge.
+ **Reviewer RpMp:** The paper addresses a timely issue—preventing unsafe content generation.

**[Novelty]:**

+ **Reviewer My5p:** The paper presents a novel and intriguing idea.
+ **Reviewer RpMp:** I appreciate the novel ideas, such as fine-tuning the safety filter within the decoder to act as a conditional decoder and the effective strategy of transforming the min-max optimization into min-min in Equation 23. Both are technically sound and effective solutions.

**[Comprehensive Analysis]:**

+ **Reviewer k1Hz:** Evaluation is conducted on multiple datasets, and multiple baseline alignment methods are compared.

During the response period, we carefully try our best to provide feedback and conduct supplementary experiments to all comments from reviewers. We concisely summarize our responses to general concerns here (For details and more questions, please refer to rebuttals below):

+ **[Evaluation]:** We conduct supplementary experiments regarding seven new evaluation metrics and one SOTA attack.
+ **[Working Scenario]:** We give detailed explanation of our working scenario, i.e., defending white-box adversary, by comparing our defense with the moderator-based baselines and other alignment-based baselines.
+ **[Method Illustration]:** We clarify some key components of our defense, e.g., Non-Fine-Tunability Enhancement, Feature Space Calibration, smoothed denial-of-service loss, simulated fine-tuning set $\phi$ by providing more intuition, detailed explanation, and relationships with existing related work.

Thank you all again for your comments, and we hope our response can address your concerns. If you have any other concerns, please let us know~

Best regards,
Author #14032

---

### Meta-Review · Area_Chair_ZmGG · 2024-12-20

**Metareview:**

This paper proposes Patronus, a defense framework for text-to-image (T2I) models designed to prevent unsafe content generation. The proposed solution embeds a safety filter directly within the decoder and employs a non-fine-tunable learning scheme, making it resistant to malicious fine-tuning. The authors validate their framework on several datasets, comparing it to baseline methods, and highlight its effectiveness in mitigating unsafe content generation while maintaining performance on benign prompts.

Overall, the paper studies a timely and significant topic of mitigating unsafe content generation in T2I models. It proposes a novel solution to the problem. Experiments include multiple datasets and adversarial attacks, demonstrating the effectiveness of the method. However, this paper has weaknesses in the lack of clarity in the experimental settings and use of the CLIP score to measure the defense capability. Those concerns have not been fully addressed after author responses.

Although the paper shows promise and addresses an important research gap, further improvement and more comprehensive evaluations are needed to strengthen its contributions and make it suitable for publication. Therefore, the AC would recommend rejection.

**Additional Comments On Reviewer Discussion:**

Reviewer My5p raised several concerns about the evaluation, including lack of evaluation results on SOTA attacks, lack of comparison with close-related baselines, limited evaluation metrics, and lack of adaptive attack evaluations. Although the authors have tried to address these concerns, the reviewer still thinks that ``the lack of clarity in the experimental settings makes the paper feel fragmented and difficult to follow`` and ``the paper requires significant reorganization, particularly in the methodology and experimental sections``.

Reviewer k1Hz raised concerns about the unclear motivation, too simple evaluation metrics, and lack of strong attacks. After author response, the reviewer still thought the concerns were not addressed.

Reviewer BgyQ initially raised concerns about computation resources. The reviewer further raised concerns about the organization of experimental results and evaluation metric during the discussion phase. As a result, the reviewer would like to downgrade the score and recommend rejection.

Reviewer RpMp initially raised several concerns about novelty, evaluation metric, and unclear strategies. The reviewer is satisfied with the author response and would recommend acceptance.

After author-reviewer discussion and AC-reviewer discussion, one reviewer would recommend acceptance and three reviewers would recommend rejection. The AC found that there are indeed some issues to the addressed and the paper needs further improvements for clarity. Therefore, the AC would recommend rejection.

---

### Decision · Program_Chairs · 2025-01-22

Reject